# Temporally Abstract Partial Models

**Khimya Khetarpal** [*1,2], **Zafarali Ahmed** [3], **Gheorghe Comanici** [3], **Doina Precup**[1,2,3]
[1]McGill University, [2]Mila, [3]DeepMind

## Abstract

Humans and animals have the ability to reason and make predictions about different courses of action at many time scales. In reinforcement learning, option models (Sutton, Precup & Singh, 1999; Precup, 2000) provide the framework for this kind of temporally abstract prediction and reasoning. Natural intelligent agents are also able to focus their attention on courses of action that are relevant or feasible in a given situation, sometimes termed affordable actions. In this paper, we define a notion of affordances for options, and develop temporally abstract partial option models, that take into account the fact that an option might be affordable only in certain situations. We analyze the trade-offs between estimation and approximation error in planning and learning when using such models, and identify some interesting special cases. Additionally, we empirically demonstrate the ability to learn both affordances and partial option models online resulting in improved sample efficiency and planning time in the Taxi domain.

## 1 Introduction

Intelligent agents flexibly reason about the applicability and effects of their actions over different time scales, which in turn allows them to consider different courses of action. Yet modeling the entire complexity of a realistic environment is quite difficult and requires a lot of data (Kakade et al., 2003). Animals and people exhibit a powerful ability to control the modelling process by understanding which actions deserve any consideration at all in a situation. By anticipating only certain aspects of their effects over different time horizons may make models more predictable or easier to learn. In this paper we develop the theoretical underpinnings of how such an ability could be defined and studied in sequential decision making. We work in the context of model-based reinforcement learning (MBRL) (Sutton and Barto, 2018) and temporal abstraction in the framework of options Sutton et al. (1999). Theories of embodied cognition and perception suggest that humans are able to represent the world knowledge in the form of *internal models* across different time scales (Pezzulo and Cisek, 2016). Option models provide a framework for RL agents to exhibit the same capability. Options define a way of behaving, including a set of states in which an option can start, an internal policy that is used to make decisions while the option is executing, and a stochastic, state-dependent termination condition. Models of options predict the (discounted) reward that an option would receive over time and the (discounted) probability distribution over the states attained at termination (Sutton et al., 1999). Consequently, option models enable the extension of dynamic programming and many other RL planning methods in order to achieve temporal abstraction, i.e. to be able to consider seamlessly different time scales of decision-making.

Much of the work on learning and planning with options considers the case where they apply everywhere (Bacon et al., 2017; Harb et al., 2017; Harutyunyan et al., 2019b,a), with some notable recent exceptions which generalize the notion of initiation sets in the context of function approximation (Khetarpal et al., 2020b). Having options that are partially defined is very important in order to control the complexity of the planning and exploration process. However, the notion of *partially defined option models*, which make predictions only from a subset of states is the focus of our paper.

---

*Correspondence to khimya.khetarpal@mail.mcgill.ca

In natural intelligence, the ability to make predictions across different scales is linked with the ability to understand the *action possibilities* (i.e. affordances) (Gibson, 1977) which arise at the interface of an agent and an environment and are a key component of successful adaptive control (Fikes et al., 1972; Korf, 1983; Drescher, 1991; Cisek and Kalaska, 2010). Recent work (Khetarpal et al., 2020a) has described a way to implement affordances in RL agents, by formalizing a notion of *intent* over state space, and then defining an affordance as the set of state-action pairs that *achieve* that intent to a certain degree. One can then plan with partial, approximate models that map affordances to intents, incurring a quantifiable amount of error at the benefit of faster learning and deliberation. In this paper, we generalize the notion of intents and affordances to option models. As we will see in Sec. 3, this is non-trivial and requires carefully inspecting the definition of option models. The resulting temporally abstract models are partial, in the sense that they apply only in certain states and options.

**Key Contributions.** We present a framework defining temporally extended intents, affordances and abstract partial option models (Sec. 3). We derive theoretical results quantifying the loss incurred when using such models for planning, exposing trade-offs between single-step models and full option models (Sec. 4). Our theoretical guarantees provide insights and decouple the role of affordances from temporal abstraction. Empirically, we demonstrate end-to-end learning of affordances and partial option models, showcasing significant improvement in final performance and sample efficiency when used for planning in the Taxi domain (Sec. 5).

## 2    Background

In RL, a decision-making agent interacts with an environment through a sequence of actions, in order to learn a way of behaving (aka policy) that maximizes its value, i.e. long-term expected return (Sutton and Barto, 2018). This process is typically formalized as a Markov Decision Process (MDP). A finite MDP is a tuple $M = \langle \mathcal{S}, \mathcal{A}, r, P, \gamma \rangle$, where $\mathcal{S}$ is a finite set of states, $\mathcal{A}$ is a finite set of actions, $r : \mathcal{S} \times \mathcal{A} \to [0, R_{\max}]$ is the reward function, $P : \mathcal{S} \times \mathcal{A} \to Dist(\mathcal{S})$ is the transition dynamics, mapping state-action pairs to a distribution over next states, and $\gamma \in [0, 1)$ is the discount factor. At each time step $t$, the agent observes a state $s_t \in \mathcal{S}$ and takes an action $a_t \in \mathcal{A}$ drawn from its policy $\pi : \mathcal{S} \to Dist(\mathcal{A})$ and, with probability $P(s_{t+1}|s_t, a_t)$, enters the next state $s_{t+1} \in \mathcal{S}$ while receiving a numerical reward $r(s_t, a_t)$. The value function of policy $\pi$ in state $s$ is the expectation of the long-term return obtained by executing $\pi$ from $s$, defined as: $V^\pi(s) = E\left[\sum_{t=0}^\infty \gamma^t r(S_t, A_t) \big| S_0 = s, A_t \sim \pi(\cdot|S_t), S_{t+1} \sim P(\cdot|S_t, A_t) \; \forall t\right]$.

The goal of the agent is to find an optimal policy, $\pi^* = \arg\max_\pi V^\pi$. If the model of the MDP, consisting of $r$ and $P$, is given, the value iteration algorithm can be used to obtain the optimal value function, $V^*$, by computing the fixed-point of the Bellman equations (Bellmann, 1957): $V^*(s) = \max_a \left( r(s, a) + \gamma \sum_{s'} P(s'|s, a) V^*(s') \right), \forall s$. The optimal policy $\pi^*$ can be obtained by acting greedily with respect to $V^*$.

**Semi-Markov Decision Process (SMDP).** An SMDP (Puterman, 1994) is a generalization of MDPs, in which the amount of time between two decision points is a random variable. The transition model of the environment is therefore a joint distribution over the next decision state and the time, conditioned on the current state and action. SMDPs obey Bellman equations similar to those for MDPs.

**Options.** Options (Sutton et al., 1999) provide a framework for temporal abstraction which builds on SMDPs, but also leverages the fact that the agent acts in an underlying MDP. A Markovian option $o$ is composed of an *intra-option policy* $\pi_o$, a termination condition $\beta_o : \mathcal{S} \to Dist(\mathcal{S})$, where $\beta_o(s)$ is the probability of terminating the option upon entering $s$, and an initiation set $I_o \subseteq \mathcal{S}$. Let $\Omega$ be the set of all options. In this document, we will use $\mathcal{O} \subset \Omega$ to denote the set of options available to the agent and $\mathcal{O}(s) = \{o | s \in I_o\}$ denote the set of options available at state $s$. In *call-and-return* option execution, when an agent is at a decision point, it examines its current state $s$, chooses $o \in \mathcal{O}(s)$ according to a policy over options $\pi_\Omega(s)$, then follows the internal policy $\pi_o$, until the option terminates according to $\beta_o$. Termination yields a new decision point, where this process is repeated.

**Option Models.** The model of an option $o$ predicts its reward and transition dynamics following a state $s \in I_o$, as follows: $r(s, o) \doteq E[R_{t+1} + \gamma R_{t+2} + \cdots + \gamma^{k-1} R_{t+k} | S_t = s, O_t = o]$, and $p(s'|s, o) \doteq \sum_{k=1}^\infty Pr(S_k = s', T_k = 1, T_{0<i<k} = 0 | S_0 = s, A_{0:k-1} \sim \pi_o, T_{0:k-1} \sim \beta_o) =$

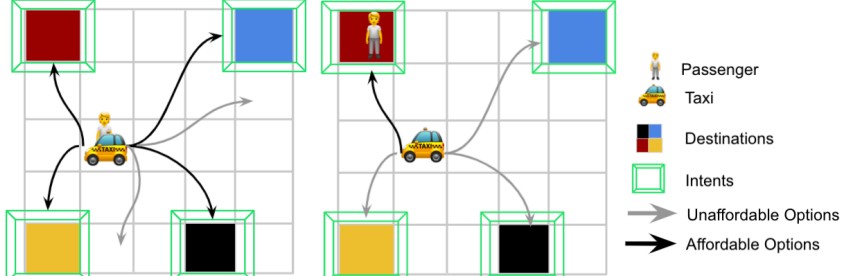

Figure 1: **Illustration:** Intents and affordances in a simple navigation task. Intents include navigation to a particular location to pick up or drop off a passenger. Affordances can indicate e.g. if a passenger can be dropped off (in the case where the passenger is already in the taxi) or if an option to pickup the passenger can succeed or fail (in the case when there is no passenger at the given location). Experiments in this domain are included in Sec. 5.

$\sum_{k=1}^{\infty} \gamma^k p(s', k|s, o)$, where $T_i$ is an indicator variable equal to $1$ if the option terminates upon entering state $i$, and $0$ otherwise. $p(s', k|s, o)$ is the probability that option $o$ terminates in $s'$ after exactly $k$ time-steps, given that it started at $s$. Bellman optimality equations can then be expressed in terms of option models. The optimal state value function and state-option value function, $V_\Omega^*$ and $Q_\Omega^*$, are defined as follows:

$$V_\Omega^*(s) = \max_{o \in \mathcal{O}(s)} Q_\Omega^*(s, o) \text{ and } Q_\Omega^*(s, o) = r(s, o) + \sum_{s'} p(s'|s, o) \max_{o' \in \mathcal{O}(s')} Q_\Omega^*(s', o').$$

**Partial Models.** MBRL methods build reward and transition models from data, which are then used to plan, e.g. by using the Bellman equations. However, learning an accurate model can be quite difficult, requiring a lot of data. Moreover, the model does not need to be accurate everywhere, as long as it is accurate in relevant places, and/or it provides useful information for identifying good actions. A useful approach is to build *partial models* (Talvitie and Singh, 2009), which only make predictions for specific parts of the observation-action space. Partial models come in two flavors: predicting only the outcome of a subset of state-action pairs, or making predictions only about certain parts of the observation space. Option models can be interpreted as partial models, of the first type, because they are defined only on states where the option applies.

**Affordances.** Gibson (1977) coined the term "affordances" to describe the fact that certain states enable certain actions, in the context of embodied agents. For instance, a chair "affords" sitting for humans, water "affords" swimming for fish, etc. As a result, affordances are a function of the environment as well as the agent, and *emerge* out of their interaction. In the context of Object Oriented-MDPs (Diuk et al., 2008), Abel et al. (2014, 2015) define affordances as propositional functions on states, which assume the existence of *objects* and *object class* descriptions. We build on a more general notion of affordances in MDPs (Khetarpal et al., 2020a), defined as a relation between states and actions, where an action is affordable in a state if its desired outcome (i.e. *intent*) is likely to be achieved.

## 3   Affordances for Temporal Abstractions

We seek to reduce both the planning complexity when using option models, and the sample complexity of learning such models, by actively eliminating from consideration choices that are unlikely to improve the planning outcome. In particular, we build temporally abstract partial models informed by affordances. Previous work (Khetarpal et al., 2020a) has formalized affordances in RL by considering the desired outcome of a primitive action, i.e. the intent associated with the action. We will now generalize this notion to intents for options, which can be achieved over the duration of the option. To make this idea concrete, consider the example of a taxicab, which needs to pick up passengers from given locations and drop them off at a desired destination. As discussed in Dietterich (2000), the use of abstraction, in both state space and time, can help solve this problem. In this context, an option could be to navigate at a particular grid location and an *intent* would be to pick up a passenger, or to drop off the passenger currently in the car at the desired destination. Such an intent limits the

space of possible options under consideration to those that have desired consequences. These intents capture long-term desired consequences of executing options.

Given the generalization of intents to temporal abstraction, the notion of affordance can still be defined similarly to the primitive action case in Khetarpal et al. (2020a), by including state-option pairs which achieve the intent to a certain degree. Indeed, primitive affordances will be a special case of option affordances. Some examples of affordances for our illustration are depicted in Fig. 1. An agent can then build partial models of only affordable options enabling it to not only *"navigate in the affordance landscape"* (Pezzulo and Cisek, 2016), but also to better gauge action choices (Cisek and Kalaska, 2010).

### 3.1 Trajectory Based Option Models

In order to justify the upcoming definitions, we will start with a slight re-writing of the option models in terms of trajectories. A trajectory $\tau(t)$ is a random variable, denoting a state-action sequence of length $t \geq 1$, $\tau(t) = \langle S_0, A_0, \ldots S_{t-1}, A_{t-1}, S_t \rangle$. Overloading notation, let $\tau(s, t)$ denote a trajectory of length $t$ for which $S_0 = s$. Further, let $\tau(s, t, s')$ be a trajectory of length $t$ with $S_0 = s$ and $S_t = s'$ and $\tau(s, s')$ a trajectory of *any length* $t$ for which $S_0 = s$ and $S_t = s'$. The return is then a deterministic function of a trajectory: $G(\tau) = \sum_{k=0}^{|\tau|-1} \gamma^k r(S_k, A_k)$, where $|\tau|$ is the length of the trajectory. The probability of observing a given trajectory $\langle s, a_0 \ldots s_t \rangle$, $s \in I_o$, under option $o$ is:

$$P(\tau = \langle s_0, a_0 \ldots s_t \rangle | o) = \left( \prod_{k=0}^{t-1} \pi_o(A_k = a_k | S_k = s_k) P(S_{k+1} = s_{k+1} | S_k = s_k, A_k = a_k)(1 - \beta_o(s_{k+1})) \right) \frac{\beta_o(s_t)}{1 - \beta_o(s_t)}$$

where the last fraction is there just to capture correctly termination at $t$. To simplify notation, we denote this by $P_o(\tau(s, t))$. We can define analogously the probability of a trajectory being generated by $o$ starting at state $s \in I_o$ and ending at a given state $s'$ after $t$ steps by $P_o(\tau(s, t, s'))$. The probability of a trajectory of any length $\tau(s, s')$ under $o$ is then: $P_o(\tau(s, s')) = \sum_{t=1}^{\infty} P_o(\tau(s, t, s'))$ Let $\mathcal{T}(s, t, s')$ denote the set of all trajectories starting at $s$, ending at $s'$ and of length $t$ and $\mathcal{T}(s, s') = \cup_t \mathcal{T}(s, t, s')$. We can write the *undiscounted transition model* of an option $o$ as:

$$P(s' | s, o) = \sum_{\tau(s, s') \in \mathcal{T}(s, s')} P_o(\tau(s, s'))$$

The discount on a trajectory $\tau$ will be denoted $\gamma(\tau)$. If the discount factor is fixed per time step, this will simply be $\gamma^{|\tau|}$; all trajectories of the same length will have the same discount, which will allow us to factor it out of products.

The *reward model* of an option is:

$$r(s, o, s') = \sum_{t=1}^{\infty} \sum_{\tau(s, t, s') \in \mathcal{T}(s, t, s')} P_o(\tau(s, t, s')) G(\tau(s, t, s'))$$

The *expected discount for option $o$ on a trajectory* going from $s$ to $s'$ is defined as:

$$\gamma_o(s, s') = \sum_{t=1}^{\infty} \sum_{\tau(s, t, s') \in \mathcal{T}(s, t, s')} P_o(\tau(s, t, s')) \gamma^t$$

Note that when the action is a primitive action, then $\gamma_o(s, s') = \sum_{s'} P(s' | s, o) \gamma$ We can re-write the optimal value function of an option as:

$$Q^*(s, o) = \sum_{s' \in \mathcal{S}} \sum_{t=1}^{\infty} \sum_{\tau(s, t, s')} P(\tau(s, t, s') | o)[G(\tau(s, t, s')) + \gamma(\tau(s, t, s')) \max_{o'} Q^*(s', o')]$$

Note that the order of the two outer sums can be reversed. This form is equivalent to the one in Sutton et al. (1999), but will be more useful for our results.

### 3.2 Option Affordances

We will now define an intent through a desired probability distribution in the space of all possible trajectories of an option. The goal will be to obtain a strict generalization of the results established in Khetarpal et al. (2020a) for primitive actions, in the case where each action is an option and $\beta(s) = 1, \forall s$.

**Definition 1** (Temporally Extended Intent $I_o^{\rightarrow}$): *A temporally extended intent of option $o \in \Omega$, $I_o^{\rightarrow} : \mathcal{S} \rightarrow Dist(\mathcal{T})$ specifies for each state $s$, a probability distribution over the space of trajectories*

$\mathcal{T}$, describing the intended result of executing $o$ in $s$. The associated intent model will be denoted by $P_I(\tau|s,o) = I_o^\rightarrow(s,\tau)$. A temporally extended intent $I_o^\rightarrow$ is satisfied to a degree, $\zeta_{s,o}$ at state $s \in \mathcal{S}$ and option $o \in \Omega$ if and only if:

$$d(P_I(\tau|s,o), P_o(\tau(s))) \leq \zeta_{s,o}, \tag{1}$$

where $d$ is a metric between probability distributions[2], and $\tau(s,o)$ denotes the trajectory starting in state $s$ and following the option $o$.

We note that primitive actions have a "degenerate" trajectory, consisting of only the next state. Hence, the only reasonable choice there is to define intent based on the next-state distribution, as done in Khetarpal et al. (2020a). However, options have a whole trajectory, and defining intents on the trajectory distribution provides maximum flexibility. In practice, we expect that most useful intents would be defined in relation with the endpoint of the option, e.g. specifying an intended distribution over the state at the end of the option, or over the joint distribution of the state and duration. Further discussion of special cases is included in the Appendix. Based on this notion of temporally extended intents, *affordances* for options can be defined as follows:

**Definition 2** (Option Affordances $\mathcal{AF}_{\mathcal{I}^\rightarrow}$): *Given a set of options $\mathcal{O} \subseteq \Omega$ and set of temporally extended intents $\mathcal{I}^\rightarrow = \cup_{o \in \mathcal{O}} I_o^\rightarrow$, and $\zeta^{\mathcal{I}^\rightarrow} \in [0,1]$, we define the affordances $\mathcal{AF}_{\mathcal{I}^\rightarrow}$ associated with $\mathcal{I}^\rightarrow$ as a relation $\mathcal{AF}_{\mathcal{I}^\rightarrow} \subseteq \mathcal{S} \times \mathcal{O}$, such that $\forall (s,o) \in \mathcal{AF}_{\mathcal{I}^\rightarrow}, I_o^\rightarrow$ is satisfied to at $(s,o)$ to degree $\zeta_{s,o} \leq \zeta^{\mathcal{I}^\rightarrow}$.*

Intuitively, we specify temporally extended intents such as "pick up passenger", "drop a passenger at destination", etc. such that the intent is satisfied to a certain degree. Affordances can then be defined as the subset of state-option pairs that can satisfy the intent to a that degree. Fig. 1 depicts a cartoon illustration of intents and corresponding option affordances in the classic Taxi environment.

## 4 Theoretical Analysis

We now analyze the value loss (Sec. 4.1) and planning loss (Sec. 4.2) induced by *temporally extended intents* $\mathcal{I}^\rightarrow$ and corresponding *temporally abstract affordances* $\mathcal{AF}_{\mathcal{I}^\rightarrow}$.

**Lemma 1.** *Given a finite set of option $\mathcal{O} \subset \Omega$ and a set of temporally extended intents $\mathcal{I}^\rightarrow = \cup_{o \in \mathcal{O}} I_o^\rightarrow$ that are satisfied to degrees $\zeta_{s,o}$, there exist constants $(\zeta_P^{\mathcal{I}^\rightarrow}, \zeta_R^{\mathcal{I}^\rightarrow})$, such that:*

$$\max_{s,o,t,s'} \sum_{\tau(s,t,s') \in \mathcal{T}(s,t,s')} \left| P_o(\tau(s,t,s')) - P_I(\tau(s,t,s')|s,o)) \right| \leq \zeta_P^{\mathcal{I}^\rightarrow} \text{ and} \tag{2}$$

$$\max_{s,o} \left| r(s,o) - E_{\tau \sim P_I}[G(\tau|s,o)] \right| \leq \zeta_R^{\mathcal{I}^\rightarrow} \tag{3}$$

*where $\zeta_P^{\mathcal{I}^\rightarrow} := \max_{s,o} \zeta_{s,o}$, $\zeta_R^{\mathcal{I}^\rightarrow} := \zeta_P^{\mathcal{I}^\rightarrow} ||G||_\infty$, and $G(\tau)$ is the return on the trajectory $\tau$.*

The proof is in the Appendix A.1.1. We note that the error in the approximate probability distribution is bounded by the degree of intent satisfaction for each option i.e $\zeta_{s,o}$. If intents are far from the true distribution $P$ (i.e. much larger $d$ in Def. 1) or misspecified, then the bounds above are predominantly governed by the approximation error induced due to the intent specification. Moreover, the approximate reward distribution is also a factor of the error in approximating probability distribution.

### 4.1 Value Loss Bound

A set of *temporally extended intents* $\mathcal{I}^\rightarrow$ define an intent-induced SMDP $\mathcal{M}_{\mathcal{I}^\rightarrow}$, in which the intents can be used to approximate the option transition and reward models. The lemma above establishes this approximation, which in turn allows us to compute the value loss incurred when planning in the intent-induced SMDP.

**Theorem 1** (Trajectory-Based Value-Loss Bound). *Given a SMDP $\mathcal{M}$ corresponding to a finite set of options $\mathcal{O}$ and a set of temporally extended intents $\mathcal{I}^\rightarrow = \cup_{o \in \mathcal{O}} I_o^\rightarrow$ defined on option trajectories (Def. 1), the value loss between the optimal policy for the original SMDP $\mathcal{M}$ and the optimal policy*

---

[2]In this work, we use $d$ to be the total variation.

$\pi^*_{\mathcal{I}^\rightarrow}$ *for the induced SMDP* $\mathcal{M}_{\mathcal{I}^\rightarrow}$ *is given by:*

$$\left\|V^{\pi^*_{\mathcal{I}^\rightarrow}} - V^*\right\|_\infty \leq \frac{\zeta_R^{\mathcal{I}^\rightarrow}}{\left(1 - \gamma^{\mathcal{I}^\rightarrow}\right)} + \frac{2R_{max}^{\mathcal{O}} \sum_{t=1}^\infty \gamma^t |\mathcal{S}| \zeta_P^{\mathcal{I}^\rightarrow}}{\left(1 - \gamma^{\mathcal{I}^\rightarrow}\right)\left(1 - \gamma^{\mathcal{O}}\right)} \tag{4}$$

*where* $\zeta_P^{\mathcal{I}^\rightarrow}$ *and* $\zeta_R^{\mathcal{I}^\rightarrow}$ *are defined in Lemma 1,* $R_{max}^{\mathcal{O}} = \max_{s,o} r(s, o)$ *is the maximum option reward,* $\gamma^{\mathcal{I}^\rightarrow} = \max_{s,o} \sum_{s'} \gamma_o^I(s, s')$ *and* $\gamma^{\mathcal{O}} = \max_{s,o} \sum_{s'} \gamma_o(s, s')$ *are the maximum expected discount factor for the intents and options respectively.*

Proof is in Appendix A.2.1. Our result is a strict generalization of the results established for primitive actions (Khetarpal et al., 2020a). Note that the value loss bound is better for temporally extended options than for primitives, due to the dependence on the maximum expected option discount (See Table 1). Note that in our bounds, $R_{max}^{\mathcal{O}}$ and $R_{max}$ denote the maximum achievable reward for options and primitive actions respectively. Further interesting corollaries are included in the Appendix.

| | Value Loss Bound | |
|---|---|---|
| **Actions** | **Sub-probability Intent** | **Trajectory based Intent** |
| **Primitive** | $2\zeta^{\mathcal{I}} \frac{\gamma R_{max}}{(1-\gamma)^2}$ | - |
| **Temporally Extended** | $2\zeta^{\mathcal{I}^\rightarrow} \frac{\gamma R_{max}^{\mathcal{O}}}{(1-\gamma)^2}$ | $\frac{\zeta_R^{\mathcal{I}^\rightarrow}}{\left(1-\gamma^{\mathcal{I}^\rightarrow}\right)} + \frac{2R_{max}^{\mathcal{O}} \sum_{t=1}^\infty \gamma^t |\mathcal{S}| \zeta_P^{\mathcal{I}^\rightarrow}}{\left(1-\gamma^{\mathcal{I}^\rightarrow}\right)\left(1-\gamma^{\mathcal{O}}\right)}$ |

Table 1: **Value Loss Analysis.** The maximum value loss incurred when considering intents shows that while both primitive ($\mathcal{I}$) and temporally extended intents ($\mathcal{I}^\rightarrow$) predominantly depend on the intent approximation error $\zeta$, temporally extended intents can result in gains contingent on the closeness of the intent model and maximum expected discounting of options and intents.

## 4.2 Planning Loss Bound

In this section, we analyze the effect of incorporating affordances and use temporally extended intents to build partial option models from data on the speed of planning. Similar results have previously been established to spell out the role of the planning horizon (Jiang et al., 2015) and to plan affordance-based partial models of primitive actions (Khetarpal et al., 2020a).

In practical scenarios, the agent may have limited information about the true model of the world. Moreover, it might be infeasible and intractable to build a full model, especially in real-life applications. To address this, we consider the SMDP $\mathcal{M}_{\mathcal{I}^\rightarrow}$ induced by models associated with temporally extended intents and the associated affordances, and quantify the loss incurred when planning with this model.

**Theorem 2** (Trajectory-Based Planning-Loss Bound). *Let* $\mathcal{I}^\rightarrow$ *be a set of temporally extended intents for a finite set of options* $\mathcal{O}$, *and* $\hat{M}_{\mathcal{AF}_{\mathcal{I}^\rightarrow}}$ *the corresponding approximate SMDP over affordable state-option pairs* $\mathcal{AF}_{\mathcal{I}^\rightarrow}$. *Then, the loss incurred when using* $\hat{M}_{\mathcal{AF}_{\mathcal{I}^\rightarrow}}$ *to compute a policy* $\pi^*_{\hat{\mathcal{M}}_{\mathcal{AF}_{\mathcal{I}^\rightarrow}}}$ *and then using this policy in the original MDP* $\mathcal{M}$ *(also known as the certainty-equivalence planning loss) can be bounded by:*

$$\left\|V^* - V^{\pi^*_{\hat{\mathcal{M}}_{\mathcal{AF}_{\mathcal{I}^\rightarrow}}}}\right\|_\infty \leq \frac{5\zeta_R^{\mathcal{I}^\rightarrow}}{(1-\gamma^{\mathcal{I}^\rightarrow})} + \frac{2R_{max}^{\mathcal{O}}}{(1-\gamma^{\mathcal{I}^\rightarrow})(1-\gamma^{\mathcal{O}})}\left(2\sum_{t=1}^\infty \gamma^t |\mathcal{S}| \zeta_P^{\mathcal{I}^\rightarrow} + \sqrt{\frac{1}{2n}\log\frac{2|\mathcal{AF}_{\mathcal{I}^\rightarrow}||\Pi_{\mathcal{I}^\rightarrow}|}{\delta}}\right)$$

*with probability at least* $1 - \delta$, *where* $\zeta_P^{\mathcal{I}^\rightarrow}$ *and* $\zeta_R^{\mathcal{I}^\rightarrow}$ *are defined in Lemma 1,* $R_{max}^{\mathcal{O}} = \max_{s,o} r(s, o)$ *is the maximum option reward,* $\gamma^{\mathcal{I}^\rightarrow} = \max_{s,o} \sum_{s'} \gamma_o^I(s, s')$ *and* $\gamma^{\mathcal{O}} = \max_{s,o} \sum_{s'} \gamma_o(s, s')$ *are the maximum expected discount factor for the intents and options respectively.*

The proof is in Appendix A.3.1. The planning loss result generalizes the result for primitive actions provided in Khetarpal et al. (2020a). We note a similar effect of incorporating affordances in partial models for temporally extended actions. The accuracy in approximation of the intent (via $(\zeta_P^{\mathcal{I}^\rightarrow}, \zeta_R^{\mathcal{I}^\rightarrow})$), the size of affordable state-option pairs $|\mathcal{AF}_{\mathcal{I}^\rightarrow}|$, and the SMDP policy class $\Pi_{\mathcal{I}}^\rightarrow$ will induce a trade-off between approximation of the intents and space of affordances. A key difference

| | Planning Loss Bound | |
| :---: | :---: | :---: |
| **Actions** | **Without Affordances** | **Affordance-aware** |
| **Primitive** | $\frac{2R_{max}}{(1-\gamma)^2} \times \left( \sqrt{\frac{1}{2n} \log \frac{2|S||A||\Pi_{S\times A}|}{\delta}} \right)$ | $\frac{2R_{max}}{(1-\gamma)^2} \times \left( 2\gamma\zeta^{\mathcal{I}} + \sqrt{\frac{1}{2n} \log \frac{2|\mathcal{AF}_{\mathcal{I}}||\Pi_{\mathcal{I}}|}{\delta}} \right)$ |
| **TEA** | $\frac{2R_{max}^{\mathcal{O}}}{(1-\gamma)^2} \left( \sqrt{\frac{1}{2n} \log \frac{2|S||\mathcal{O}||\Pi_{S\times\mathcal{O}}|}{\delta}} \right)$ | $\frac{2R_{max}^{\mathcal{O}}}{(1-\gamma)^2} \left( 2\gamma\zeta^{\mathcal{I}^{\rightarrow}} + \sqrt{\frac{1}{2n} \log \frac{2|\mathcal{AF}_{\mathcal{I}^{\rightarrow}}||\Pi_{\mathcal{I}^{\rightarrow}}|}{\delta}} \right)$ |

Table 2: **On the role of affordances in actions and options.** We decouple the role of the temporal extent of the options and the effects of incorporating affordances. Our analysis establishes improved guarantees for planning with option models. Further gains are obtained when affordances are incorporated, though at the cost of increased approximation error due to intents through $\zeta$. We note that for simplicity, we present the bounds obtained when intents are defined on the distribution of an option's terminal state, a corollary of Theorem 2. The table highlights the trade-offs between *estimation* (via the model learning depending on the data size $n$) and *approximation* (via the specification of intents).

in planning with the approximate partial option models $\hat{M}_{\mathcal{AF}_{\mathcal{I}^{\rightarrow}}}$ is that the error can be controlled through the maximum expected discount factor for both intent and option models which in turn depends on the minimum expected duration of all affordable options.

Table 2 summarizes the effects of using temporally extended models and affordances. First, we note that the planning with affordances introduces a trade-off between *approximation* and *estimation* in both primitive and temporally extended actions. Concretely, the approximation error is induced due to the specification of intents through $\zeta^{\mathcal{I}^{\rightarrow}}$, whereas the estimation error is induced due to learning of the transition and has a dependence on the data size $n$ and the size of the policy class $\Pi_{\mathcal{I}^{\rightarrow}}$.

## 5   Empirical Analysis

In this section, we study the impact of using affordances to learn partial option models which are then used for planning, in order to corroborate the theoretical results established in Sec. 4. In Sec. 5.1, we use a hand designed set of affordances to show that it can improve training stability as well as sample efficiency when used to learn a single partial option model, conditioned on a state-option pair. Then, in Sec. 5.2 we demonstrate the viability of learning the set of affordances at the same time as the partial option model resulting in a set of affordances that were smaller than those that were hand designed.

**Environment.** We consider the $5 \times 5$ Taxi domain (Dietterich, 2000). The domain is a grid world with four designated pickup/drop locations, marked as R(ed), B(lue), G(reen), and Y(ellow). See Fig. 1 for illustration. The agent controls a taxi and faces an episodic problem: the taxi starts in a randomly-chosen square and is given a goal location at which a passenger must be dropped. The passenger is at one of the three other locations. To complete the task, the agent must drive the taxi to the passenger's location, pick them up, go to the destination, and drop the passenger there. The action space consists of six primitive actions: Up, Down, Left, Right, Pickup, and Drop. The agent gets a reward of $-1$ per step, $+20$ for successfully dropping the passenger at the goal and $-10$ for dropping the passenger at the wrong location. There are a total of 25 (grid positions) $\times 4$ (goal destinations) $\times 5$ (passenger scenarios) $= 500$ states in this environment and the observation is a one-hot vector.

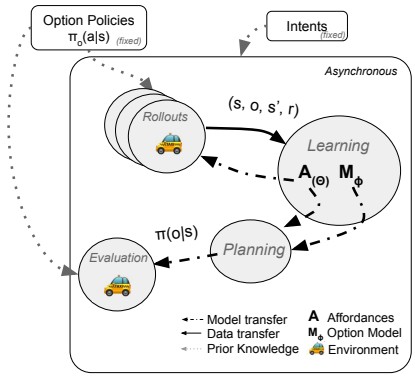

Figure 2: **Experimental pipeline.**

**Option set $\mathcal{O}$.** We consider a fixed set of *taxi-centric* options, defined as follows: Go to a grid position (25 options); Drop passenger at grid position (25 options); Pickup passenger from grid position (25 options). The options are pre-trained via value iteration and fixed for all our experiments. In total there are $75 \times 500 = 37500$ state-option pairs.

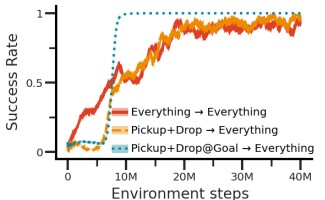
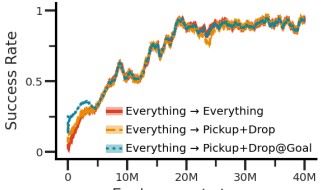
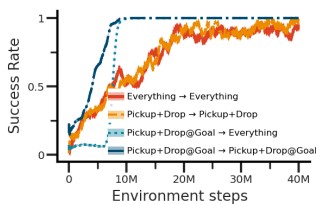

(a) Data collection and model learning with affordances.

(b) Planning with affordances.

(c) Data collection, model learning and planning with affordances.

Figure 3: **The impact of affordance sets on success rate at different parts of the learning pipeline.** (a) The use of affordances improves model learning even in the absence of any affordances during planning (blue dotted). (b) The use of affordances did not impact planning because the underlying quality of the model is the same. (c) When using affordances both during model learning and planning (blue dashed), the best performance is obtained. Curves are smoothed over 4 independent seeds using ggplot's `stat_smooth` using a span of 0.1 and confidence interval of 95%.

**Experimental pipeline.** [3] We use pre-trained options, $o = \langle I_o = \mathcal{S}, \pi_o(a|s), \beta_o(s) \rangle$, to collect transition data $(s_t, o, T, s_{t+T}, r = \sum_{i=t}^{T} r_i)$ where option $o$ was initiated at state $s_t$ and ended in state $s_{t+T}$ after $T$ steps, accumulating a reward of $r$. We execute options until termination or for $T_{\max}$ steps, whichever comes first. We learn linear models to predict the next state distribution $\hat{P}_{\phi_1}(s'|s, o)$, option duration, $\hat{L}_{\phi_2}(s, o)$ and reward $\hat{r}_{\phi_3}(s, o)$, where $\phi$ denote parameter vectors. Affordances can be incorporated in model learning by selecting only affordable options during the data collection and to mask the loss of unaffordable state-option transitions:

$$\sum_{(s,o,T,s',r) \in \mathcal{D}} A(s, o, s', \mathcal{I}^{\rightarrow}) \left[ -\log \hat{P}_{\phi_1}(s'|o, s) + (\hat{L}_{\phi_2}(o, s) - T)^2 + (\hat{r}_{\phi_3}(s, o) - r)^2 \right] \quad (5)$$

where $A(s, o, s', I)$ is 1 if $(s, o, s')$ is affordable according to the intent $I$ and 0 otherwise. We use the learned models, $\hat{M}$, in value iteration to obtain a policy over options $\pi_{\mathcal{O}}(o|s_t)$. Affordances can be incorporated into planning by only considering state-option pairs in the affordance set (See Algorithm 1 in the Appendix). We report the *success rate*, i.e., the proportion of episodes in which the agent successfully drops the passenger at the correct location. Data collection, learning, and evaluation happen asynchronously and simultaneously (Fig 2) using the Launchpad framework (Yang et al., 2021).

### 5.1 Intents and affordances are most useful in model learning when the affordance sets are more relevant.

In this section we investigate the utility of using affordances on different aspects of the pipeline by considering a fixed set of affordances used either during model learning or planning. We first define three intent sets, $\mathcal{I}^{\rightarrow}$, and their corresponding affordances:

1. **Everything**: All options are affordable at every state resulting in 37,500 state-option pairs in this affordance set.
2. **Pickup+Drop**: We build this set of affordances heuristically, by eliminating all options that simply go to a grid position, resulting in 25,000 state-option pairs .
3. **Pickup+Drop@Goal**: We create this affordance set of 4,000 state-option pairs that terminate at the four destination positions only.

When learning the partial model, using the most restrictive and relevant affordance set (**Pickup+Drop@Goal**) to collect data and mask the loss (*→ Everything) significantly improves the sample efficiency (Fig. 3(a)). The difference between **Everything** and **Pickup+Drop** was insignificant suggesting that the order of magnitude decrease in the number of state-option pairs in the affordance set is important (See also Sec 5.2 for more analysis of the affordance set size). Additionally, using any affordance set enables the use of a higher learning rate for learning the model without divergence (Fig. B1). On the other hand, given the same option model, using affordance sets only during planning (Everything→*) does not create any improvement in the success rate (Fig. 3(b)) or decrease in the planning iterations (Fig. 4): the quality of the model dictates the success rate.

---

[3] We will provide the source code for our empirical analysis here.

Finally, using the most restrictive affordance set for both model learning and planning (Pickup+Drop@Goal→Pickup+Drop@Goal) can result in further improvements in the sample efficiency (Fig. 3(c)) as well as accelerated planning time (Fig. 4)) demonstrating a combined benefit of using affordances in more aspects of the pipeline.

## 5.2 Relevant affordances can be learned online and result in improved sample efficiency.

In this section, we demonstrate the ability to learn affordances at the same time as learning the partial option model. To do this, we train a classifier, $A_\theta(s, o, s', I) \in [0, 1]$ corresponding to intent $I \in \mathcal{I}^\rightarrow$, which predicts if a state-option pair is affordable. **Pickup+Drop@Goal** is defined by 8 intents: four that are completed when the agent has a passenger in the vehicle at the destinations; and four that are completed when the agent has dropped the passenger at the destinations. We convert $A_\theta(s, o, s', I)$ into an indicator for Eq. 5, by ensuring that at least one of the intents in the intent set is affordable, $A(s, o, s', \mathcal{I}^\rightarrow) = \mathbb{1}[(\max_{I \in \mathcal{I}^\rightarrow}(A(s, o, s', I)) > k]$ at some threshold value, $k$. When $k = 0$, all state and options are affordable. The affordance classifier is learned at the same time as the option model, $\hat{M}$, using the standard cross entropy objective:

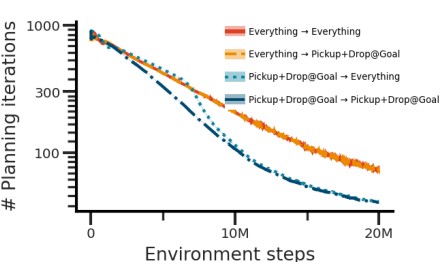

Figure 4: **Improvements in planning iterations when using affordances**. When using affordances during model learning and in both model learning and planning, we get sustained decrease in planning iterations compared to not using them or only using them during planning.

$-\sum_{I \in \mathcal{I}^\rightarrow} c(s, o, s', I) \log A(s, o, s', I)$ where $c(s, o, s', I)$ is the intent completion function indicating if intent $I$ was completed during the transition.

The threshold, $k$, controls the size of the affordance set (Fig. 5(a)) with larger $k$'s resulting in smaller affordance sets. The learned affordance set for **Pickup+Drop@Goal** is 2,000 state-option pairs which smaller than what we heuristically defined (4,000 state-option pairs). Smaller affordance sets result in improved sample efficiency (Fig. 5(b)). We highlight that this is not necessarily obvious since the learned affordance sets could remove potentially useful state-options pairs and $k$ would be used to control how restrictive the sets are. These results show that affordances can be learned online for a defined set of intents and result in good performance. In particular, there are sample efficiency gains by using more restricted affordance sets.

Our results here demonstrate empirically that learning a partial option model requires much fewer samples as opposed to learning a full model. We also corroborate this with theoretical guarantees on sample and computational complexity of obtaining an $\varepsilon$-estimation of the optimal option value function, given only access to a generative model (See Appendix Sec. C).

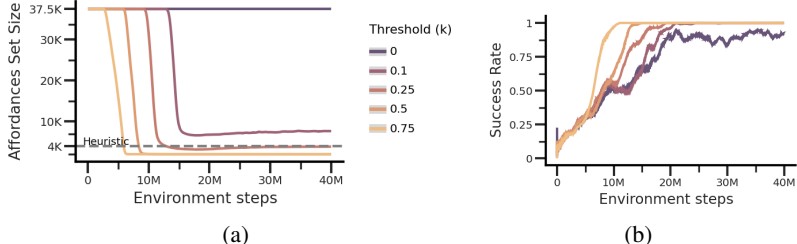

Figure 5: **The impact of learning the affordance set for Pickup+Drop@Goal on (a) size of the affordance set and (b) success in the downstream task.** There is a one-to-one correspondence between the threshold, $k$, the affordance set size and the success rate on the taxi task. The learned affordance set for Pickup+Drop@Goal is smaller than the heuristic used in Fig. 3(c).

# 6 Related Work

Affordances are viewed as the action opportunities (Gibson, 1977; Chemero, 2003), emerging out of the agent-environment interaction (Heft, 1989), and have been typically studied in AI as possibilities associated with an object (Slocum et al., 2000; Fitzpatrick et al., 2003; Lopes et al., 2007; Montesano et al., 2008; Cruz et al., 2016, 2018; Fulda et al., 2017; Song et al., 2015; Abel et al., 2014). Affordances have also been formalized in RL without the assumption of objects (Khetarpal et al., 2020a). Our work presents the general case of temporal abstraction (Sutton et al., 1999).

The process model of behavior and cognition (Pezzulo and Cisek, 2016) in the space of affordances is expressed at multiple levels of abstraction. During interactive behavior, action representations at different levels of abstraction can indeed be mapped to findings about the way in which the human brain adaptively selects among predictions of outcomes at different time scales (Cisek and Kalaska, 2010; Pezzulo and Cisek, 2016).

In RL, the generalization of one-step action models to option models (Sutton et al., 1999) enables an agent to predict and reason at multiple time scales. Precup et al. (1998) established dynamic programming results for option models which enjoy similar theoretical guarantees as primitive action models. Abel et al. (2019) proposed expected-length models of options. Our theoretical results can also be extended to expected-length option models.

Building agents that can represent and use predictive knowledge requires efficient solutions to cope with the combinatorial explosion of possibilities, especially in large environments. Partial models (Talvitie and Singh, 2009) provide an elegant solution to this problem, as they only model part of the observation. Existing methods focus on predictions for only some of the observations (Oh et al., 2017; Amos et al., 2018; Guo et al., 2018; Gregor et al., 2019; Zhao et al., 2021), but they still model the effects of all actions and focus on single-step dynamics (Watters et al., 2019). Recent work by Xu et al. (2020) proposed a deep RL approach to learn partial models with goals akin to intents, which is complementary to our work.

# 7 Conclusions and Limitations

We presented notions of intents and affordances that can be used together with options. They allow us to define *temporally abstract partial models*, which extend option models to be conditioned on affordances. Our theoretical analysis suggests that modelling temporally extended dynamics for only relevant parts of the environment-agent interface provides two-fold benefits: 1) faster planning across different timescales (Sec. 4), and 2) improved sampled efficiency (Appendix Sec. C). However, these benefits can come at the cost of some increase in approximation bias, but this tradeoff can still be favourable. For example, in the low-data regime, intermediate-size affordances (much smaller than the entire state-option space) could really improve the speed of planning. Picking intents judiciously can also induce sample complexity gains, if the approximation error due to the intent is manageable. Our empirical illustration shows that our approach can produce significant benefits.

**Limitations & Future Work.** Our analysis assumes that the intents and options are fixed apriori. To learn intents, we envisage an iterative algorithm which alternates between learning intents and affordances, such that intents can be refined over time and the mis-specifications can also be self-corrected (Talvitie, 2017). Our analysis is complimentary to any method for providing or discovering intents. Another important future direction is to build partial option models and leverage their predictions in large scale problems (Vinyals et al., 2019). Besides, it would be useful to relate our work to cognitive science models of *intentional options*, which can reason about the space of future affordances (Pezzulo and Cisek, 2016). Aligned with future affordances, a promising research avenue is to study the emergence of *new* affordances at the boundary of the agent-environment interaction in the presence of non-stationarity (Chandak et al., 2020).

## Acknowledgments and Disclosure of Funding

The authors would like to thank Feryal Behbahani and Dave Abel for a very detailed feedback, Martin Klissarov and Emmanuel Bengio for valuable comments on a draft of this paper, and Joelle Pineau for feedback on ideas presented in this work. A special thank you to Ahmed Touati for discussion and detailed notes (Azar et al., 2012) presented in RL theory reading group at Mila.

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
