## A Proofs

### A.1 Lemmas and Remarks

#### A.1.1 Proof of Lemma 1

*Proof.* (Approximate Probability Distributions) From Def. 1, $\forall I_o^{\rightarrow} \in \mathcal{I}^{\rightarrow}$, $I_o^{\rightarrow}$ is satisfied to a degree, $\zeta_{s,o}$ at state $s \in \mathcal{S}$ and option $o \in \mathcal{O}$ if and only if:

$$d(P_I(\tau|s, o), P_o(\tau(s))) \leq \zeta_{s,o},$$

where $d$ is a metric between probability distributions. Let $\zeta_P^{\mathcal{I}^{\rightarrow}} = \max_{s,o} \zeta_{s,o}$. The result follows immediately.

(Approximate Reward Distributions) Let $\zeta_R^{\mathcal{I}^{\rightarrow}} = \left|\left|G\right|\right|_{\infty} \zeta_P^{\mathcal{I}^{\rightarrow}}$. We now consider the maximum error in approximation of rewards due to intent specification as follows:

$$\max_{s,o} \left| r(s, o) - E_{\tau \sim P_I}[G(\tau|s, o)] \right|$$

$$= \max_{s,o} \left| \sum_{s'} r(s, o, s') - \sum_{s'} \sum_{\tau} \sum_{t=1}^{\infty} P_I(\tau(s, t, s')|s, o))G(\tau(s, t, s')) \right|$$

$$= \max_{s,o} \left| \sum_{s'} \sum_{\tau} \sum_{t=1}^{\infty} P_o(\tau(s, t, s')|s, o))G(\tau(s, t, s'))- \right.$$

$$\left. \sum_{s'} \sum_{\tau} \sum_{t=1}^{\infty} P_I(\tau(s, t, s')|s, o))G(\tau(s, t, s')) \right|$$

$$= \max_{s,o} \left| \sum_{s'} \sum_{t=1}^{\infty} \left( \sum_{\tau} P_o(\tau(s, t, s')|s, o)) - P_I(\tau(s, t, s')|s, o)) \right) G(\tau(s, t, s')) \right|$$

$$\leq \left|\left|G\right|\right|_{\infty} \zeta_P^{\mathcal{I}^{\rightarrow}} = \zeta_R^{\mathcal{I}^{\rightarrow}}$$

$\square$

#### A.1.2 Remarks

**Remark 1.** *Given a finite SMDP $\mathcal{M}$, a finite set of options $\mathcal{O}$, the maximum achievable optimal value function $\left|\left|V^*\right|\right|_{\infty}$ is upper bounded by $\frac{R_{max}^{\mathcal{O}}}{(1-\gamma^{\mathcal{O}})}$ where $\gamma^{\mathcal{O}} = \max_{s,o} \sum_{s'} \gamma_o(s, s')$, and $R_{max}^{\mathcal{O}} = \max_{s,o} R(s, o)$.*

*Proof.* To upper bound the optimal value function, we consider $\left|\left|Q^*\right|\right|_{\infty} = \max_{s,o} Q^*(s, o) = \max_s \underbrace{\max_o Q^*(s, o)}_{V^*}$. Then, $\forall\, s, o \in \mathcal{S}, \mathcal{O}$ :

$$Q^*(s, o) = \sum_{s'} \sum_{t=1}^{\infty} \sum_{\tau(s,t,s')} P(\tau(s, t, s')|o)[G(\tau(s, t, s') + \gamma(\tau(s, t, s')) \max_{o'} Q^*(s', o')]$$

$$= R(s, o) + \sum_{s'} \sum_{t=1}^{\infty} \sum_{\tau(s,t,s')} P(\tau(s, t, s')|o)\gamma(\tau(s, t, s')) \max_{o'} Q^*(s', o')$$

Taking the max norm on both sides,

$$\max_{s,o} Q^*(s,o) = \left\|R(s,o) + \sum_{s'}\sum_{t=1}^{\infty}\sum_{\tau(s,t,s')} P(\tau(s,t,s')|o)\gamma(\tau(s,t,s'))\max_{o'} Q^*(s',o')\right\|_{\infty}$$

$$\leq \max_{s,o} R(s,o) + \max_{s,o}\underbrace{\sum_{s'}\sum_{t=1}^{\infty}\sum_{\tau(s,t,s')} P(\tau(s,t,s')|o)\gamma(\tau(s,t,s'))\max_{o'} Q^*(s',o')}_{\sum_{s'}\gamma_o(s,s')}$$

$$\leq R_{max}^{\mathcal{O}} + \left\|Q^*\right\|_{\infty}\max_{s,o}\sum_{s'}\gamma_o(s,s')$$

$$\implies \left\|Q^*\right\|_{\infty} \leq R_{max}^{\mathcal{O}}\left(1 - \max_{s,o}\sum_{s'}\gamma_o(s,s')\right)^{-1}$$

$$\implies \left\|V^*\right\|_{\infty} \leq R_{max}^{\mathcal{O}}\left(1 - \max_{s,o}\sum_{s'}\gamma_o(s,s')\right)^{-1} = R_{max}^{\mathcal{O}}\left(1 - \gamma^{\mathcal{O}}\right)^{-1}.$$

$\square$

**Remark 2.** *Given a finite SMDP $\mathcal{M}$, a finite set of options $\mathcal{O}$, with $\mathcal{D}$ as the minimum expected duration for which all options execute, $\gamma$ to be the maximum expected option discount factor, the maximum achievable optimal value function $V_{max}$ is upper bounded by $\frac{R_{max}^{\mathcal{O}}}{(1-\gamma^{\mathcal{D}})} = \frac{R_{max}^{\mathcal{O}}}{(1-\gamma^{\mathcal{O}})}$, where $R_{max}^{\mathcal{O}}$ is the maximum achievable reward by an option, and $\mathcal{D} = \min_{s,o}\log_{\gamma}\sum_{s'} p(s'|s,o)$.*

*Proof.* Consider the maximum achievable optimal value function in the SMDP $\mathcal{M}$ to be $V_{max}$.

$$V_{max} = ||V_{\mathcal{O}}^*||_{\infty}.$$

Then, $\forall\, s \in \mathcal{S}$:

$$V_{\mathcal{O}}^*(s) = \max_{o\in\mathcal{O}}\left[R(s,o) + \sum_{s'} p(s'|s,o)V_{\mathcal{O}}^*(s')\right]$$

$$\leq \max_{o\in\mathcal{O}}\left[R(s,o) + \sum_{s'} p(s'|s,o)\max_{s''\in\mathcal{S}} V_{\mathcal{O}}^*(s'')\right]$$

$$= \max_{o\in\mathcal{O}}\left[R(s,o) + \gamma_o(s)\max_{s''\in\mathcal{S}} V_{\mathcal{O}}^*(s'')\right], \quad \text{substituting } \gamma_o(s) = \sum_{s'} p(s'|s,o)$$

$$= \max_{o\in\mathcal{O}}\left[R(s,o) + \gamma_o(s)||V_{\mathcal{O}}^*||_{\infty}\right]$$

$$\leq \underbrace{\max_{o\in\mathcal{O}} R(s,o)}_{\leq R_{max}^{\mathcal{O}}} + \underbrace{\max_{s,o\in\mathcal{S},\mathcal{O}}\gamma_o(s)}_{\leq \gamma_{max}}||V_{\mathcal{O}}^*||_{\infty}$$

$$\leq R_{max}^{\mathcal{O}} + \gamma_{max}||V_{\mathcal{O}}^*||_{\infty}, \quad \text{where } R_{max}^{\mathcal{O}} = \max_{s,o} R(s,o)$$

$$\implies ||V_{\mathcal{O}}^*||_{\infty} \leq \frac{R_{max}^{\mathcal{O}}}{(1 - \gamma_{max})}.$$

Note consider the following definition of $\mathcal{D}$:

$$\mathcal{D} = \min_{s,o}\log_{\gamma}\sum_{s'} p(s'|s,o) = \min_{s,o}\log_{\gamma}\gamma_o(s)$$

$$= \log_{\gamma}\underbrace{\max_{s,o}\gamma_o(s)}_{\gamma_{max}}, \quad \text{since } \gamma < 1, \log_{\gamma} \text{ is a monotonically decreasing function}$$

$$= \log_{\gamma}\gamma_{max}$$

$$\implies \gamma_{max} = \gamma^{\mathcal{D}} \implies \gamma^{\mathcal{D}} = \gamma^{\mathcal{O}}$$

Therefore, $V_{max} \leq \frac{R_{max}^{\mathcal{O}}}{(1-\gamma^{\mathcal{O}})}$.

$\square$

## A.2 Proofs - Value Loss Analysis

**Note:** For convenience, throughout our proofs we will be using $\mathcal{I}$ instead of $\mathcal{I}^{\rightarrow}$ to denote a set of temporally extended intents. Similarly, we will use $I$ instead of $\mathrm{I}_o^{\rightarrow}$ to denote a temporally extended intent for an option $o$.

### A.2.1 Proof of Theorem 1

*Proof.* Formally, the value loss is defined as

$$\left\|V_{\mathcal{M}}^{\pi_{\mathcal{I}}^*} - V_{\mathcal{M}}^*\right\|_\infty = \max_{s \in \mathcal{S}} \left|V_{\mathcal{M}}^{\pi_{\mathcal{I}}^*}(s) - V_{\mathcal{M}}^*(s)\right|$$

We now consider the RHS and expand as follows:

$$\max_{s \in \mathcal{S}} \left|V_{\mathcal{M}}^{\pi_{\mathcal{I}}^*}(s) - V_{\mathcal{M}}^*(s)\right| \leq \underbrace{\max_{s \in \mathcal{S}} \left|V_{\mathcal{M}}^*(s) - V_{\mathcal{M}_{\mathcal{I}}}^*(s)\right|}_{\textbf{Term 1}} + \underbrace{\max_{s \in \mathcal{S}} \left|V_{\mathcal{M}}^{\pi_{\mathcal{I}}^*}(s) - V_{\mathcal{M}_{\mathcal{I}}}^*(s)\right|}_{\textbf{Term 2}}$$

*Bounding Term 1.*

$$\max_{s \in \mathcal{S}} \left|V_{\mathcal{M}}^*(s) - V_{\mathcal{M}_{\mathcal{I}}}^*(s)\right| = \max_{s \in \mathcal{S}} \max_{o \in \mathcal{O}} \left|Q^*(s, o) - Q_I^*(s, o)\right|$$

Expanding the action-value loss from the RHS above, we get:

$$Q^*(s, o) - Q_I^*(s, o) =$$

$$= \sum_{s'} \sum_{t=1}^{\infty} \sum_{\tau(s,t,s')} P(\tau(s,t,s')|o)[G(\tau(s,t,s')) + \gamma(\tau(s,t,s')) \max_{o'} Q^*(s', o')]$$

$$- \sum_{s'} \sum_{t=1}^{\infty} \sum_{\tau(s,t,s')} P_I(\tau(s,t,s')|o)[G(\tau(s,t,s')) + \gamma(\tau(s,t,s')) \max_{o'} Q_I^*(s', o')]$$

$$= \sum_{s'} \sum_{t=1}^{\infty} \sum_{\tau(s,t,s')} (P(\tau(s,t,s')|o) - P_I(\tau(s,t,s')|o)G(\tau(s,t,s'))$$

$$+ \sum_{s'} \sum_{t=1}^{\infty} \sum_{\tau(s,t,s')} \gamma(\tau(s,t,s'))\Big(P(\tau(s,t,s')|o) \max_{o'} Q^*(s', o') - P_I(\tau(s,t,s')|o) \max_{o'} Q_I^*(s', o')\Big)$$

$$= (R(s, o) - R_I(s, o)) + \sum_{s'} \sum_{t=1}^{\infty} \sum_{\tau(s,t,s')} \gamma(\tau(s,t,s'))\Big(P(\tau(s,t,s')|o) \max_{o'} Q^*(s', o')$$

$$- P_I(\tau(s,t,s')|o) \max_{o'} Q^*(s', o') + P_I(\tau(s,t,s')|o) \max_{o'} Q^*(s', o') - P_I(\tau(s,t,s')|o) \max_{o'} Q_I^*(s', o')\Big)$$

$$= (R(s, o) - R_I(s, o)) + \sum_{s'} \sum_{t=1}^{\infty} \sum_{\tau(s,t,s')} \gamma(\tau(s,t,s'))(P(\tau(s,t,s')|o) - P_I(\tau(s,t,s')|o)) \max_{o'} Q^*(s', o')$$

$$+ \sum_{s'} \sum_{t=1}^{\infty} \sum_{\tau(s,t,s')} \gamma(\tau(s,t,s'))P_I(\tau(s,t,s')|o)) \max_{o'}(Q^*(s', o') - Q_I^*(s', o'))$$

Taking the max norm and applying triangle inequality, we get:

$$\left\|Q^* - Q_I^*\right\|_\infty = \max_{s,o}\Big[(R(s,o) - R_I(s,o))+$$

$$\sum_{s'}\sum_{t=1}^\infty\sum_{\tau(s,t,s')}\gamma(\tau(s,t,s'))(P(\tau(s,t,s')|o) - P_I(\tau(s,t,s')|o))\max_{o'}Q^*(s',o')$$

$$+\sum_{s'}\sum_{t=1}^\infty\sum_{\tau(s,t,s')}\gamma(\tau(s,t,s'))P_I(\tau(s,t,s')|o))\max_{o'}(Q^*(s',o') - Q_I^*(s',o'))\Big]$$

$$\leq\left\|R - R_I\right\|_\infty + \max_{s,o}\sum_{s'}\sum_{t=1}^\infty\sum_{\tau(s,t,s')}\gamma(\tau(s,t,s'))\Big(P(\tau(s,t,s')|o) - P_I(\tau(s,t,s')|o)\Big)\|Q^*\|_\infty$$

$$+\underbrace{\max_{s,o}\sum_{s'}\sum_{t=1}^\infty\sum_{\tau(s,t,s')}\gamma(\tau(s,t,s'))P_I(\tau(s,t,s')|o))}_{\sum_{s'}\gamma_o^I(s,s')}\left\|Q^* - Q_I^*\right\|_\infty$$

$$\leq\left\|R - R_I\right\|_\infty + \max_{s,o}\sum_{s'}\sum_{t=1}^\infty\sum_{\tau(s,t,s')}\gamma(\tau(s,t,s'))\Big(P(\tau(s,t,s')|o) - P_I(\tau(s,t,s')|o)\Big)\left\|Q^*\right\|_\infty$$

$$+\max_{s,o}\sum_{s'}\gamma_o^I(s,s')\left\|Q^* - Q_I^*\right\|_\infty$$

Rearranging, we get:

$$\left\|Q^* - Q_I^*\right\|_\infty \leq \Big(1 - \max_{s,o}\sum_{s'}\gamma_o^I(s,s')\Big)^{-1}\Big[\left\|R - R_I\right\|_\infty +$$

$$\max_{s,o}\sum_{s'}\sum_{t=1}^\infty\sum_{\tau(s,t,s')}\gamma(\tau(s,t,s'))\big(P(\tau(s,t,s')|o) - P_I(\tau(s,t,s')|o))\big)\left\|Q^*\right\|_\infty\Big]$$

Since $V^*(s) = \max_o Q^*(s,o)$, we can rewrite the above as following:

$$\left\|V_{\mathcal{M}}^* - V_{\mathcal{M}_I}^*\right\|_\infty \leq \Big(1 - \max_{s,o}\sum_{s'}\gamma_o^I(s,s')\Big)^{-1}\Big[\left\|R - R_I\right\|_\infty +$$

$$+\max_{s,o}\sum_{s'}\sum_{t=1}^\infty\sum_{\tau(s,t,s')}\gamma(\tau(s,t,s'))\big(P(\tau(s,t,s')|o) - P_I(\tau(s,t,s')|o))\big)\left\|V^*\right\|_\infty\Big]$$

*Bounding Term 2.* We now consider the term 2 and bound the policy evaluation error i.e. $\max_{s \in \mathcal{S}} \left| V_{\mathcal{M}}^{\pi_{\mathcal{I}}^*}(s) - V_{\mathcal{M}_{\mathcal{I}}}^{\pi_{\mathcal{I}}^*}(s) \right|$

$$V_{\mathcal{M}}^{\pi_{\mathcal{I}}^*}(s) - V_{\mathcal{M}_{\mathcal{I}}}^{\pi_{\mathcal{I}}^*}(s) =$$

$$= \sum_{s'} \sum_{t=1}^{\infty} \sum_{\tau(s,t,s')} P(\tau(s,t,s')|\pi_{\mathcal{I}}^*(s))[G(\tau(s,t,s')) + \gamma(\tau(s,t,s'))V_{\mathcal{M}}^{\pi_{\mathcal{I}}^*}(s')]$$

$$- \sum_{s'} \sum_{t=1}^{\infty} \sum_{\tau(s,t,s')} P_I(\tau(s,t,s')|\pi_{\mathcal{I}}^*(s))[G(\tau(s,t,s')) + \gamma(\tau(s,t,s'))V_{\mathcal{M}_{\mathcal{I}}}^{\pi_{\mathcal{I}}^*}(s')]$$

$$= \sum_{s'} \sum_{t=1}^{\infty} \sum_{\tau(s,t,s')} \Big( P(\tau(s,t,s')|\pi_{\mathcal{I}}^*(s)) - P_I(\tau(s,t,s')|\pi_{\mathcal{I}}^*(s)) \Big) G(\tau(s,t,s'))$$

$$+ \sum_{s'} \sum_{t=1}^{\infty} \sum_{\tau(s,t,s')} \gamma(\tau(s,t,s')) \Big( P(\tau(s,t,s')|\pi_{\mathcal{I}}^*(s)) V_{\mathcal{M}}^{\pi_{\mathcal{I}}^*}(s') - P_I(\tau(s,t,s')|\pi_{\mathcal{I}}^*(s)) V_{\mathcal{M}_{\mathcal{I}}}^{\pi_{\mathcal{I}}^*}(s') \Big)$$

$$= (R(s, \pi_{\mathcal{I}}^*(s)) - R_I(s, \pi_{\mathcal{I}}^*(s))) + \sum_{s'} \sum_{t=1}^{\infty} \sum_{\tau(s,t,s')} \gamma(\tau(s,t,s')) \Big( P(\tau(s,t,s')|\pi_{\mathcal{I}}^*(s)) V_{\mathcal{M}}^{\pi_{\mathcal{I}}^*}(s')$$

$$- P_I(\tau(s,t,s')|\pi_{\mathcal{I}}^*(s)) V_{\mathcal{M}}^{\pi_{\mathcal{I}}^*}(s') + P_I(\tau(s,t,s')|\pi_{\mathcal{I}}^*(s)) V_{\mathcal{M}}^{\pi_{\mathcal{I}}^*}(s') - P_I(\tau(s,t,s')|\pi_{\mathcal{I}}^*(s)) V_{\mathcal{M}_{\mathcal{I}}}^{\pi_{\mathcal{I}}^*}(s') \Big)$$

$$= (R(s, \pi_{\mathcal{I}}^*(s)) - R_I(s, \pi_{\mathcal{I}}^*(s))) + \sum_{s'} \sum_{t=1}^{\infty} \sum_{\tau(s,t,s')} \gamma(\tau(s,t,s'))(P(\tau(s,t,s')|\pi_{\mathcal{I}}^*(s)) - P_I(\tau(s,t,s')|\pi_{\mathcal{I}}^*(s))) V_{\mathcal{M}}^{\pi_{\mathcal{I}}^*}(s')$$

$$+ \sum_{s'} \sum_{t=1}^{\infty} \sum_{\tau(s,t,s')} \gamma(\tau(s,t,s')) P_I(\tau(s,t,s')|\pi_{\mathcal{I}}^*(s)) \Big( V_{M}^{\pi_{\mathcal{I}}^*}(s') - V_{\mathcal{M}_{\mathcal{I}}}^{\pi_{\mathcal{I}}^*}(s') \Big)$$

Taking the max over all states, and applying triangle inequality we get:

$$\max_{s \in \mathcal{S}} \left| V_{\mathcal{M}}^{\pi_{\mathcal{I}}^*}(s) - V_{\mathcal{M}_{\mathcal{I}}}^{\pi_{\mathcal{I}}^*}(s) \right| =$$

$$\max_{s} \Big| (R(s, \pi_{\mathcal{I}}^*(s)) - R_I(s, \pi_{\mathcal{I}}^*(s)))$$

$$+ \sum_{s'} \sum_{t=1}^{\infty} \sum_{\tau(s,t,s')} \gamma(\tau(s,t,s'))(P(\tau(s,t,s')|\pi_{\mathcal{I}}^*(s)) - P_I(\tau(s,t,s')|\pi_{\mathcal{I}}^*(s))) V_{\mathcal{M}}^{\pi_{\mathcal{I}}^*}(s')$$

$$+ \sum_{s'} \sum_{t=1}^{\infty} \sum_{\tau(s,t,s')} \gamma(\tau(s,t,s')) P_I(\tau(s,t,s')|\pi_{\mathcal{I}}^*(s)) \Big( V_{M}^{\pi_{\mathcal{I}}^*}(s') - V_{\mathcal{M}_{\mathcal{I}}}^{\pi_{\mathcal{I}}^*}(s') \Big) \Big|$$

$$\leq ||R - R_I||_{\infty}$$

$$+ \max_{s} \sum_{s'} \sum_{t=1}^{\infty} \sum_{\tau(s,t,s')} \gamma(\tau(s,t,s')) |P(\tau(s,t,s')|\pi_{\mathcal{I}}^*(s)) - P_I(\tau(s,t,s')|\pi_{\mathcal{I}}^*(s))| \left|\left| V_{\mathcal{M}}^{\pi_{\mathcal{I}}^*} \right|\right|_{\infty}$$

$$+ \max_{s} \sum_{s'} \sum_{t=1}^{\infty} \sum_{\tau(s,t,s')} \gamma(\tau(s,t,s')) P_I(\tau(s,t,s')|\pi_{\mathcal{I}}^*(s)) \left|\left| V_{M}^{\pi_{\mathcal{I}}^*}(s') - V_{\mathcal{M}_{\mathcal{I}}}^{\pi_{\mathcal{I}}^*} \right|\right|_{\infty}$$

Rearranging the terms, we get:

$$\left|\left| V_{\mathcal{M}}^{\pi_{\mathcal{I}}^*} - V_{\mathcal{M}_{\mathcal{I}}}^{\pi_{\mathcal{I}}^*} \right|\right|_{\infty} \leq \Big( 1 - \max_{s,o} \sum_{s'} \gamma_o^I(s,s') \Big)^{-1} \Big[ ||R - R_I||_{\infty} +$$

$$+ \max_{s,o} \sum_{s'} \sum_{t=1}^{\infty} \sum_{\tau(s,t,s')} \gamma(\tau(s,t,s')) |P(\tau(s,t,s')|o) - P_I(\tau(s,t,s')|o)|) ||V^*||_{\infty} \Big]$$

Plugging the bounds for the two terms in our original loss, and plugging the upper bound on the optimal value function from Remark 1, we get:

$$\left\|\left|V_{\mathcal{M}}^{\pi_{\mathcal{I}}^*} - V_{\mathcal{M}}^*\right|\right\|_\infty \leq \left(1 - \max_{s,o} \sum_{s'} \gamma_o^I(s,s')\right)^{-1} \left\|\left|R - R_I\right|\right\|_\infty + \frac{2R_{max}^{\mathcal{O}}\left(1 - \max_{s,o}\sum_{s'}\gamma_o^I(s,s')\right)^{-1}}{\left(1 - \max_{s,o}\sum_{s'}\gamma_o(s,s')\right)} \times$$

$$\max_{s,o} \sum_{s'} \sum_{t=1}^{\infty} \sum_{\tau(s,t,s')} \gamma(\tau(s,t,s'))\Big|P(\tau(s,t,s')|o) - P_I(\tau(s,t,s')|o))\Big|$$

Further, substituting Lemma 1, we get the final result as follows:

$$\left\|\left|V_{\mathcal{M}}^{\pi_{\mathcal{I}}^*} - V_{\mathcal{M}}^*\right|\right\|_\infty \leq \frac{\zeta_R^{\mathcal{I}}}{\left(1 - \gamma^{\mathcal{I}}\right)} + \frac{2R_{max}^{\mathcal{O}}\sum_{t=1}^{\infty}\gamma^t|\mathcal{S}|\zeta_P^{\mathcal{I}}}{\left(1 - \gamma^{\mathcal{I}}\right)\left(1 - \gamma^{\mathcal{O}}\right)}$$

Recall that $\mathcal{I}$ was used to denote $\mathcal{I}^{\rightarrow}$, the set of temporally extended intents, throughout the proof.  □

### A.2.2   Corollary 1. SMDP - Multi-Time-Model of Intent - Value Loss Bound

A special case of our formulation is to model the consequences of following a specific course of action based on final state representations at the SMDP level.

More precisely, the multi-time-model of an option intent must characterize both the target state distribution resulting upon the option's completion, and the intended temporal scale at which the option operates i.e. $I_o^{\rightarrow} : \mathcal{S} \to \text{SDist}(\mathcal{S})$, where SDist stands for the set of all sub-probability distributions over $\mathcal{S}$. The intent-induced transition model would then take the role of the transition dynamics reflected by the option model (assuming rewards are the same and known). For this case, we require a metric between sub-probability distributions and assume that,

**Assumption 1.** *For each state-option pair, the total variation between the intended distribution $P_I$ and the true distribution $P$ is bounded by a constant $\zeta_{s,o}$, i.e.*

$$\sum_{s'} \Big|P_I(s'|s,o) - p(s'|s,o)\Big| \leq \zeta_{s,o}. \tag{6}$$

*The degree of satisfaction of the intent is the maximum over all $(s,o)$ pairs, i.e. $\max_{s,o} \zeta_{s,o} = \zeta^{\mathcal{I}}$.*

**Corollary 1.** *[Multi-Time-Model of Intent- Value Loss.] Given a SMDP $\mathcal{M}$ corresponding to a set of options $\mathcal{O}$ and a set of temporally extended multi-time-model of intents, the value loss between the optimal policy for the original SMDP $\mathcal{M}$ and the optimal policy $\pi_{\mathcal{I}^\rightarrow}^*$ for the induced SMDP $\mathcal{M}_{\mathcal{I}^\rightarrow}$ is given by:*

$$\left\|\left|V_{\mathcal{M}}^{\pi_{\mathcal{I}^\rightarrow}^*} - V_{\mathcal{M}}^*\right|\right\|_\infty \leq 2\zeta^{\mathcal{I}^\rightarrow}\frac{\gamma R_{max}^{\mathcal{O}}}{(1-\gamma)^2}, \tag{7}$$

*where $\zeta^{\mathcal{I}^\rightarrow}$ is the degree of satisfaction of the intents (Eq. 6), $R_{max}^{\mathcal{O}} = \max_{s,o} r(s,o)$ is the maximum option reward, and $\gamma$ is the maximum expected option discount factor.*

*Proof.* We now show that our general result in Theorem 1 can be reduced to a specific case of considering the multi-time-option model of intents.

We first assume here that rewards are known and given which results in the term $\left\|\left|R - R_I\right|\right\|_\infty = 0$, and the second term can be simplified further as follows:

$$\left\|\left|Q^* - Q_I^*\right|\right\|_\infty \leq \frac{||Q^*||_\infty}{1 - \max_{s,o}\sum_{s'}\gamma_o^I(s,s')} \max_{s,o} \sum_{s'} \sum_{t=1}^{\infty} \sum_{\tau(s,t,s')} \gamma(\tau(s,t,s'))\Big|P(\tau(s,t,s')|o) - P_I(\tau(s,t,s')|o)\Big|$$

Plugging Remark 1, we get:

$$||V_{\mathcal{M}}^{\pi_{\mathcal{I}}^*} - V_{\mathcal{M}}^*||_\infty \leq \frac{2R_{max}^{\mathcal{O}}}{(1-\gamma)^2} \underbrace{\max_{s,o} \sum_{s'} \sum_{t=1}^{\infty} \sum_{\tau(s,t,s')} \gamma(\tau(s,t,s'))\Big|P(\tau(s,t,s')|o) - P_I(\tau(s,t,s')|o)\Big|}_{\leq \gamma\zeta^{\mathcal{I}^\rightarrow}}$$

Simplifying terms, we get the final result

$$\left\|V_{\mathcal{M}}^{\pi_{\mathcal{I}}^*} - V_{\mathcal{M}}^*\right\|_\infty \le 2\zeta^{\mathcal{I}} \frac{\gamma R_{max}^{\mathcal{O}}}{(1-\gamma)^2}$$

$\square$

## A.3 Proofs - Planning Loss Analysis

**Definition 3** (Policy class $\Pi_{\mathcal{I}\to}$): *Given affordance set $\mathcal{AF}_{\mathcal{I}\to}$, let $\mathcal{M}_{\mathcal{I}\to}$ be the set of SMDPs over the state-options pairs in $\mathcal{AF}_{\mathcal{I}\to}$, let*
$$\Pi_{\mathcal{I}\to} = \{\pi_M^*\} \cup \{\pi : \exists \bar{M} \in \mathcal{M}_{\mathcal{I}\to} \text{ s.t. } \pi \text{ is optimal in } \bar{M}\}.$$

### A.3.1 Proof of Theorem 2. Planning Loss - Trajectories Based Intent.

*Proof.* To prove this theorem we will be using the lemmas below: Lemma 2, Lemma 3, and Lemma 4, and 5.

**Note:** For convenience, throughout our proofs we will be using $\mathcal{I}$ instead of $\mathcal{I}^\to$ to denote a set of temporally extended intents. Similarly, we will use $I$ instead of $I_o^\to$ to denote a temporally extended intent for an option $o$.

**Lemma 2.** *For any SMDP $\hat{\mathcal{M}}_{\mathcal{AF}_{\mathcal{I}}}$, which is an approximate model of the SMDP given by the intent collection $\mathcal{I}$[4], we have*

$$\left\|V_{\mathcal{M}_{\mathcal{I}}}^* - V_{\mathcal{M}_{\mathcal{I}}}^{\pi_{\hat{\mathcal{M}}_{\mathcal{AF}_{\mathcal{I}}}}^*}\right\|_\infty \le 2 \max_{\pi \in \Pi_{\mathcal{I}}} \|V_{\mathcal{M}_{\mathcal{I}}}^\pi - V_{\hat{\mathcal{M}}_{\mathcal{AF}_{\mathcal{I}}}}^\pi\|_\infty. \tag{8}$$

*Proof.* $\forall s \in \mathcal{S}$, Let us consider:

$$V_{\mathcal{M}_{\mathcal{I}}}^*(s) - V_{\mathcal{M}_{\mathcal{I}}}^{\pi_{\hat{\mathcal{M}}_{\mathcal{AF}_{\mathcal{I}}}}^*}(s)$$

$$= \left(V_{\mathcal{M}_{\mathcal{I}}}^*(s) - V_{\hat{\mathcal{M}}_{\mathcal{AF}_{\mathcal{I}}}}^{\pi_{\mathcal{M}_{\mathcal{I}}}^*}(s)\right) + \underbrace{\left(V_{\hat{\mathcal{M}}_{\mathcal{AF}_{\mathcal{I}}}}^{\pi_{\mathcal{M}_{\mathcal{I}}}^*}(s) - V_{\hat{\mathcal{M}}_{\mathcal{AF}_{\mathcal{I}}}}^*(s)\right)}_{\le 0} + \left(V_{\hat{\mathcal{M}}_{\mathcal{AF}_{\mathcal{I}}}}^*(s) - V_{\mathcal{M}_{\mathcal{I}}}^{\pi_{\hat{\mathcal{M}}_{\mathcal{AF}_{\mathcal{I}}}}^*}(s)\right)$$

$$\le \left(V_{\mathcal{M}_{\mathcal{I}}}^*(s) - V_{\hat{\mathcal{M}}_{\mathcal{AF}_{\mathcal{I}}}}^{\pi_{\mathcal{M}_{\mathcal{I}}}^*}(s)\right) - \left(V_{\hat{\mathcal{M}}_{\mathcal{AF}_{\mathcal{I}}}}^*(s) - V_{\mathcal{M}_{\mathcal{I}}}^{\pi_{\hat{\mathcal{M}}_{\mathcal{AF}_{\mathcal{I}}}}^*}(s)\right)$$

$$\le 2 \max_{\pi \in \left\{\pi_{\hat{\mathcal{M}}_{\mathcal{AF}_{\mathcal{I}}}}^*, \pi_{\mathcal{M}_{\mathcal{I}}}^*\right\}} \left|V_{\mathcal{M}_{\mathcal{I}}}^\pi(s) - V_{\hat{\mathcal{M}}_{\mathcal{AF}_{\mathcal{I}}}}^\pi(s)\right|$$

Taking a max over all states on both sides of the inequality and noticing that the set of all policies is a trivial super set of $\left\{\pi_{\hat{\mathcal{M}}_{\mathcal{AF}_{\mathcal{I}}}}^*, \pi_{\mathcal{M}_{\mathcal{I}}}^*\right\}$, we get the equation in Lemma 2 above. Moreover since, our definition of $\Pi_{\mathcal{I}}$ is a superset with the optimal policies included, we can further say the following:

$$\left\|V_{\mathcal{M}_{\mathcal{I}}}^* - V_{\mathcal{M}_{\mathcal{I}}}^{\pi_{\hat{\mathcal{M}}_{\mathcal{AF}_{\mathcal{I}}}}^*}\right\|_\infty \le 2 \max_{\pi \in \Pi_{\mathcal{I}}} \|V_{\mathcal{M}_{\mathcal{I}}}^\pi - V_{\hat{\mathcal{M}}_{\mathcal{AF}_{\mathcal{I}}}}^\pi\|_\infty.$$

$\square$

**Lemma 3.** *For any SMDP $\hat{\mathcal{M}}_{\mathcal{AF}_{\mathcal{I}}}$ bounded by $[0, R_{max}^{\mathcal{O}}]$ with corresponding value function bounded by $V_{max}$ which is an approximate of the SMDP estimated from data experienced in the world for a set of intents $\mathcal{I}$,*

$$\left\|V_{\mathcal{M}_{\mathcal{I}}}^\pi - V_{\hat{\mathcal{M}}_{\mathcal{AF}_{\mathcal{I}}}}^\pi\right\|_\infty \le \frac{1}{\left(1-\gamma^{\mathcal{I}}\right)} \max_{s,o} \left|(\hat{R}_I(s,o) + \langle\hat{\gamma}(s,o,;)\hat{P}_I(s,o,;), V_{\mathcal{M}_{\mathcal{I}}}^\pi\rangle) - V_{\mathcal{M}_{\mathcal{I}}}^\pi\right|. \tag{9}$$

---

[4]We overload notation and throughout our proofs, for convenience we interchangeably use $\mathcal{I}$ and $\mathcal{I}$ to denote set of temporally extended intents.

*Proof.* Given any policy over options $\pi$, define state-value function $V_0, V_1, \ldots V_m$ such that $V_0 = V^\pi_{\mathcal{M}_\mathcal{I}}$,

From this point onward, we use $\mathcal{AF}_\mathcal{I}(o)$ and $\mathcal{AF}_\mathcal{I}(s)$ to denote affordable states and affordable options respectively. Recall that $\mathcal{AF}_\mathcal{I} \subseteq \mathcal{S} \times \mathcal{O}$.

$\forall s \in \mathcal{AF}_\mathcal{I}(o)$,

$$V_m(s) = \sum_{o \in \mathcal{AF}_\mathcal{I}(s)} \pi(o|s)\Big(\hat{R}(s,o) + \langle \hat{P}_I(s,o,;), V_{m-1}\rangle\Big)$$

Now, rewriting the above in new format:

$$V_m(s) = \sum_o \pi(o|s)\left[\sum_{s'}\sum_{t=1}^\infty \sum_{\tau(s,t,s')} \hat{P}_I(\tau(s,t,s')|o)[G(\tau(s,t,s')) + \gamma(\tau(s,t,s'))V_{m-1}(s')]\right]$$

Therefore:

$$||V_m - V_{m-1}||_\infty = \max_s \left[\sum_{o \in \mathcal{AF}_\mathcal{I}(s)} \pi(o|s)\sum_{s'}\sum_{t=1}^\infty \sum_{\tau(s,t,s')} \hat{P}_I(\tau(s,t,s')|o)\gamma(\tau(s,t,s'))(V_{m-1}(s') - V_{m-2}(s'))\right]$$

$$\leq \max_s \sum_{o \in \mathcal{AF}_\mathcal{I}(s)} \pi(o|s)\sum_{s'}\sum_{t=1}^\infty \sum_{\tau(s,t,s')} \hat{P}_I(\tau(s,t,s')|o)\gamma(\tau(s,t,s'))||V_{m-1} - V_{m-2}||_\infty$$

$$= \max_s \sum_{o \in \mathcal{AF}_\mathcal{I}(s)} \pi(o|s)\sum_{s'} \gamma^I_o(s,s')||V_{m-1} - V_{m-2}||_\infty \tag{10}$$

Since $\mathrm{E}[\sum_{s'} \gamma^I_o(s,s')] \leq \max_{s,o}\sum_{s'}\gamma^I_o(s,s')$, therefore

$$||V_m - V_{m-1}||_\infty \leq \underbrace{\max_{s,o}\sum_{s'}\gamma^I_o(s,s')}_{\gamma^\mathcal{I}} ||V_{m-1} - V_{m-2}||_\infty$$

Therefore,

$$||V_m - V_0||_\infty \sum_{k=0}^{m-1}||V_{k+1} - V_k||_\infty \leq ||V_1 - V_0||_\infty \sum_{k=1}^{m-1}(\gamma^\mathcal{I})^{k-1}.$$

Taking the limit $m \to \infty$, $V_m \to V^\pi_{\hat{\mathcal{M}}_{\mathcal{AF}_\mathcal{I}}}$, we have:

$$||V_{\hat{\mathcal{M}}_{\mathcal{AF}_\mathcal{I}}} - V_0||_\infty \leq \frac{1}{\left(1 - \gamma^\mathcal{I}\right)}||V_1 - V_0||_\infty$$

where notice that $V_0 = V^\pi_{\mathcal{M}_\mathcal{I}}$ and

$$V_1 = \sum_{o \in \mathcal{AF}_\mathcal{I}(s)} \pi(o|s)\Big(\hat{R}_I + \langle \gamma(s,o,;)\hat{P}_I(s,o;), V^\pi_M\rangle\Big).$$

Therefore,

$$\left|\left|V^\pi_{\mathcal{M}_\mathcal{I}} - V^\pi_{\hat{\mathcal{M}}_{\mathcal{AF}_\mathcal{I}}}\right|\right|_\infty$$

$$\leq \frac{1}{\left(1 - \gamma^\mathcal{I}\right)} \max_s \left|\sum_{o \in \mathcal{AF}_\mathcal{I}(s)} \pi(o|s)(\hat{R}_I(s,o) + \langle \gamma(s,o,;)\hat{P}_I(s,o,;), V^\pi_{\mathcal{M}_\mathcal{I}}\rangle) - V^\pi_{\mathcal{M}_\mathcal{I}}\right|$$

$$\leq \frac{1}{\left(1 - \gamma^\mathcal{I}\right)} \max_{s,o} \left|(\hat{R}_I(s,o) + \langle \gamma(s,o,;)\hat{P}_I(s,o,;), V^\pi_{\mathcal{M}_\mathcal{I}}\rangle) - V^\pi_{\mathcal{M}_\mathcal{I}}\right|.$$

$\square$

*Next, we turn to Lemma 4.*

**Lemma 4.** *For any SMDP $\hat{\mathcal{M}}_{\mathcal{AF}_{\mathcal{I}}}$ with value function bounded by $V_{max}$ which is an approximate of the SMDP estimated from data experienced in the world for a set of intents $\mathcal{I}$, The following holds with probability at least $1 - \delta$:*

$$\left\| V^*_{\mathcal{M}_{\mathcal{I}}} - V^{\pi^*_{\hat{\mathcal{M}}_{\mathcal{AF}_{\mathcal{I}}}}}_{\mathcal{M}_{\mathcal{I}}} \right\|_\infty \leq \frac{2R^{\mathcal{O}}_{max}}{\left(1 - \gamma^{\mathcal{I}}\right)\left(1 - \gamma^{\mathcal{O}}\right)} \sqrt{\frac{1}{2n} \log \frac{2|\mathcal{AF}_{\mathcal{I}}||\Pi_{\mathcal{I}}|}{\delta}}.$$

*Proof.* Using Lemma 2 (L2) and Lemma 3 ( L3), we have

$$\left\| V^{\pi^*_{\mathcal{M}_{\mathcal{I}}}}_{\mathcal{M}_{\mathcal{I}}} - V^{\pi^*_{\hat{\mathcal{M}}_{\mathcal{AF}_{\mathcal{I}}}}}_{\mathcal{M}_{\mathcal{I}}} \right\|_\infty \leq 2 \max_{\pi \in \Pi_{\mathcal{I}}} \left\| V^\pi_{\mathcal{M}_{\mathcal{I}}} - V^\pi_{\hat{\mathcal{M}}_{\mathcal{AF}_{\mathcal{I}}}} \right\|_\infty \; \text{L2.}$$

$$\leq \frac{2}{\left(1 - \gamma^{\mathcal{I}}\right)} \max_{\substack{\pi \in \Pi_{\mathcal{I}} \\ s \times o \in \mathcal{AF}_{\mathcal{I}}}} \left| (\hat{R}_I(s, o) + \langle \gamma(s, o, ; )\hat{P}_I(s, o, ; ), V^\pi_{\mathcal{M}_{\mathcal{I}}}\rangle) - V^\pi_{\mathcal{M}_{\mathcal{I}}} \right| \text{L3.}$$

Since $(\hat{P}_I(s, o, ; ), V^\pi_{\mathcal{M}_{\mathcal{I}}}) - V^\pi_{\mathcal{M}_{\mathcal{I}}})$ is the average of the IID samples the agent obtains by interacting with the environment, bounded in $[0, V_{max}]$ with mean $V^\pi_{\mathcal{M}_{\mathcal{I}}}$ (for any $s, o, \pi$ tuple i.e. state, option and policy over options tuple). Then according to Hoeffdings inequality,

$$\forall t \geq 0, \; P\left(\left| \sum_{o \in \mathcal{AF}_{\mathcal{I}}(s)} (\hat{R}_I(s, o) + \langle \gamma(s, o, ; )\hat{P}_I(s, o, ; ), V^\pi_{\mathcal{M}_{\mathcal{I}}}\rangle) - V^\pi_{\mathcal{M}_{\mathcal{I}}} \right| > t\right) \leq 2 \exp\left\{\frac{-2nt^2}{(V_{max})^2}\right\}$$

To obtain a uniform bound over all $s, o, \pi$ tuples, we equate the RHS to $\frac{\delta}{|\mathcal{AF}_{\mathcal{I}}(o)||\mathcal{AF}_{\mathcal{I}}(s)|\Pi_{\mathcal{I}}|}$ and the result follows as shown below.

$$2 \exp\left\{\frac{-2nt^2}{(V_{max})^2}\right\} = \frac{\delta}{|\mathcal{AF}_{\mathcal{I}}(o)||\mathcal{AF}_{\mathcal{I}}(s)||\Pi_{\mathcal{I}}|}$$

$$\frac{-2nt^2}{(V_{max})^2} = \log \frac{\delta}{2|\mathcal{AF}_{\mathcal{I}}(o)||\mathcal{AF}_{\mathcal{I}}(s)||\Pi_{\mathcal{I}}|}$$

$$\frac{2nt^2}{(V_{max})^2} = \log \frac{2|\mathcal{AF}_{\mathcal{I}}(o)||\mathcal{AF}_{\mathcal{I}}(s)||\Pi_{\mathcal{I}}|}{\delta}$$

$$t^2 = V_{max} \frac{1}{2n} \log \frac{2|\mathcal{AF}_{\mathcal{I}}(o)||\mathcal{AF}_{\mathcal{I}}(s)||\Pi_{\mathcal{I}}|}{\delta}$$

$$t = V_{max} \sqrt{\frac{1}{2n} \log \frac{2|\mathcal{AF}_{\mathcal{I}}(o||\mathcal{AF}_{\mathcal{I}}(s)||\Pi_{\mathcal{I}}|}{\delta}}$$

We express the state-option pairs in affordances as the size of affordances. Formally, the size of affordances for a intent can be expressed as $|\mathcal{AF}_{\mathcal{I}}|$. Plugging this back, and using Remark 1, we get the final result. $\qquad\square$

**Lemma 5.** *Given any policy over options $\pi$, we have*

$$\left\| V^\pi_{\mathcal{M}} - V^\pi_{\mathcal{M}_{\mathcal{I}}} \right\|_\infty \leq \frac{1}{(1 - \gamma^{\mathcal{I}})} \left(2\zeta^{\mathcal{I}}_R + \left\| V^\pi_{\mathcal{M}} \right\|_\infty \max_{s,o} \sum_{t=1}^{\infty} \gamma^t |\mathcal{S}|\zeta^{\mathcal{I}}_P\right) \qquad (11)$$

*Proof.* We will use the following Bellman operator:

$$\mathcal{T}^\pi_{\mathcal{M}}f = \sum_o \pi(o|s)\left[\sum_{s'} \sum_{t=1}^{\infty} \sum_{\tau(s,t,s')} P(\tau(s, t, s')|o)[G(\tau(s, t, s')) + \gamma(\tau(s, t, s'))f(s')]\right]$$

$$(\mathcal{T}^\pi_{\mathcal{M}_1} - \mathcal{T}^\pi_{\mathcal{M}_2})f(s)$$

$$= \sum_o \pi(o|s)\Big[\Big(R_1(s,o) - R_2(s,o)\Big) + \sum_{s'}\sum_{t=1}^\infty \sum_{\tau(s,t,s')} \gamma^t f(s')\Big(P_1(\tau(s,t,s')|o) - P_2(\tau(s,t,s')|o)\Big)\Big]$$

$$= \sum_o \pi(o|s)\Big(R_1(s,o) - R_2(s,o)\Big) + \sum_o \pi(o|s)\sum_{s'}\sum_{t=1}^\infty \gamma^t \sum_{\tau(s,t,s')} f(s')\Big(P_1(\tau(s,t,s')|o) - P_2(\tau(s,t,s')|o)\Big)$$

$$\leq \zeta_R^\mathcal{I} + \Big|\Big|f\Big|\Big|_\infty \max_{s,o}\sum_{t=1}^\infty \gamma^t \sum_{s'}\sum_{\tau(s,t,s')}\Big(P_1(\tau(s,t,s')|o) - P_2(\tau(s,t,s')|o)\Big)$$

$$\leq \zeta_R^\mathcal{I} + \Big|\Big|f\Big|\Big|_\infty \max_{s,o}\sum_{t=1}^\infty \gamma^t \sum_{s'}\Big[\sum_{\tau(s,t,s')}\Big(P_1(\tau(s,t,s')|o) - P_2(\tau(s,t,s')|o)\Big)\Big]$$

$$\leq \zeta_R^\mathcal{I} + \Big|\Big|f\Big|\Big|_\infty \sum_{t=1}^\infty \gamma^t |\mathcal{S}|\zeta_P^\mathcal{I}$$

and

$$\mathcal{T}^\pi_\mathcal{M} f_1(s) - \mathcal{T}^\pi_\mathcal{M} f_2(s) =$$

$$= \sum_o \pi(o|s)\Big(R_1(s,o) - R_2(s,o)\Big) + \sum_o \pi(o|s)\sum_{s'}\sum_{t=1}^\infty \sum_{\tau(s,t,s')} \gamma^t P_\mathcal{M}(\tau(s,t,s'))\Big(f_1(s') - f_2(s')\Big)$$

$$\leq \zeta_R^\mathcal{I} + \Big|\Big|f_1 - f_2\Big|\Big|_\infty \max_{s,o}\sum_{s'} \gamma_o^\mathcal{M}(s,s')$$

Now, the following holds for the initial value error we are interested to bound:

$$||V^\pi_\mathcal{M} - V^\pi_{\mathcal{M}_\mathcal{I}}||_\infty \leq ||V^\pi_\mathcal{M} - \mathcal{T}^\pi_{\mathcal{M}_\mathcal{I}}V^\pi_\mathcal{M}||_\infty + ||\mathcal{T}^\pi_{\mathcal{M}_\mathcal{I}}V^\pi_\mathcal{M} - V^\pi_{\mathcal{M}_\mathcal{I}}||_\infty$$

$$= ||\mathcal{T}^\pi_\mathcal{M}V^\pi_\mathcal{M} - \mathcal{T}^\pi_{\mathcal{M}_\mathcal{I}}V^\pi_\mathcal{M}||_\infty + ||\mathcal{T}^\pi_{\mathcal{M}_\mathcal{I}}V^\pi_\mathcal{M} - \mathcal{T}^\pi_{\mathcal{M}_\mathcal{I}}V^\pi_{\mathcal{M}_\mathcal{I}}||_\infty$$

$$= ||(\mathcal{T}^\pi_\mathcal{M} - \mathcal{T}^\pi_{\mathcal{M}_\mathcal{I}})V^\pi_\mathcal{M}||_\infty + ||\mathcal{T}^\pi_{\mathcal{M}_\mathcal{I}}(V^\pi_\mathcal{M} - V^\pi_{\mathcal{M}_\mathcal{I}})||_\infty$$

$$\leq \zeta_R^\mathcal{I} + \Big|\Big|V^\pi_\mathcal{M}\Big|\Big|_\infty \max_{s,o}\sum_{t=1}^\infty \gamma^t |\mathcal{S}|\zeta_P^\mathcal{I} + \zeta_R^\mathcal{I} + \max_{s,o}\sum_{s'} \gamma_o^I(s,s')||V^\pi_\mathcal{M} - V^\pi_{\mathcal{M}_\mathcal{I}}||_\infty$$

Unfolding the above to infinity, we obtain in the limit the following:

$$||V^\pi_\mathcal{M} - V^\pi_{\mathcal{M}_\mathcal{I}}||_\infty \leq \frac{1}{(1 - \max_{s,o}\sum_{s'}\gamma_o^I(s,s'))}\Big(2\zeta_R^\mathcal{I} + \Big|\Big|V^\pi_\mathcal{M}\Big|\Big|_\infty \max_{s,o}\sum_{t=1}^\infty \gamma^t |\mathcal{S}|\zeta_P^\mathcal{I}\Big)$$

Therefore,

$$||V^\pi_\mathcal{M} - V^\pi_{\mathcal{M}_\mathcal{I}}||_\infty \leq \frac{1}{(1 - \gamma^\mathcal{I})}\Big(2\zeta_R^\mathcal{I} + \Big|\Big|V^\pi_\mathcal{M}\Big|\Big|_\infty \max_{s,o}\sum_{t=1}^\infty \gamma^t |\mathcal{S}|\zeta_P^\mathcal{I}\Big)$$

$\square$

***Plugging Lemmas Back.*** *Now the following holds for the original LHS of the planning loss bound we are after.*

$$\Big|\Big|V^*_\mathcal{M} - V^{\pi^*_{\tilde{\mathcal{M}}_{\mathcal{AF}_\mathcal{I}}}}_\mathcal{M}\Big|\Big|_\infty \leq \Big|\Big|V^*_\mathcal{M} - V^{\pi^*_{\mathcal{M}_\mathcal{I}}}_\mathcal{M}\Big|\Big|_\infty + \Big|\Big|V^{\pi^*_{\mathcal{M}_\mathcal{I}}}_\mathcal{M} - V^*_{\mathcal{M}_\mathcal{I}}\Big|\Big|_\infty +$$

$$\Big|\Big|V^*_{\mathcal{M}_\mathcal{I}} - V^{\pi^*_{\tilde{\mathcal{M}}_{\mathcal{AF}_\mathcal{I}}}}_{\mathcal{M}_\mathcal{I}}\Big|\Big|_\infty + \Big|\Big|V^{\pi^*_{\tilde{\mathcal{M}}_{\mathcal{AF}_\mathcal{I}}}}_{\mathcal{M}_\mathcal{I}} - V^{\pi^*_{\tilde{\mathcal{M}}_{\mathcal{AF}_\mathcal{I}}}}_\mathcal{M}\Big|\Big|_\infty$$

*Theorem 1 applies to the first term, Lemma 5 to the second and forth term, and Lemma 4 for the third term. Finally,*

$$\left\|V_{\mathcal{M}}^* - V_{\mathcal{M}}^{\pi^*_{\hat{\mathcal{M}}_{\mathcal{AF}_{\mathcal{I}}}}}\right\|_\infty \leq \frac{1}{\left(1-\gamma^{\mathcal{I}}\right)}\zeta_R^{\mathcal{I}} + \frac{2R_{max}^{\mathcal{O}}}{\left(1-\gamma^{\mathcal{I}}\right)\left(1-\gamma^{\mathcal{O}}\right)}\max_{s,o}\sum_{t=1}^{\infty}\gamma^t|\mathcal{S}|\zeta_P^{\mathcal{I}}+$$

$$\frac{2}{(1-\gamma^{\mathcal{I}})}\left(2\zeta_R^{\mathcal{I}} + \frac{R_{max}^{\mathcal{O}}}{(1-\gamma^{\mathcal{O}})}\max_{s,o}\sum_{t=1}^{\infty}\gamma^t|\mathcal{S}|\zeta_P^{\mathcal{I}}\right)+$$

$$\frac{2R_{max}^{\mathcal{O}}}{\left(1-\gamma^{\mathcal{I}}\right)\left(1-\gamma^{\mathcal{O}}\right)}\sqrt{\frac{1}{2n}\log\frac{2|\mathcal{AF}_{\mathcal{I}}||\Pi_{\mathcal{I}}|}{\delta}}$$

*Rearranging terms, we get:*

$$\left\|V_{\mathcal{M}}^* - V_{\mathcal{M}}^{\pi^*_{\hat{\mathcal{M}}_{\mathcal{AF}_{\mathcal{I}}}}}\right\|_\infty \leq \frac{5\zeta_R^{\mathcal{I}}}{\left(1-\gamma^{\mathcal{I}}\right)} + \frac{2R_{max}^{\mathcal{O}}}{\left(1-\gamma^{\mathcal{I}}\right)\left(1-\gamma^{\mathcal{O}}\right)}\left(2\max_{s,o}\sum_{t=1}^{\infty}\gamma^t|\mathcal{S}|\zeta_P^{\mathcal{I}} + \sqrt{\frac{1}{2n}\log\frac{2|\mathcal{AF}_{\mathcal{I}}||\Pi_{\mathcal{I}}|}{\delta}}\right)$$

$$\square$$

### A.3.2 Corollary 3. SMDP - Multi-Time-Model of Intent : Planning Loss

Analogous to the value loss analysis, we obtain the special case of planning loss bound for multi-time-model of an option intent as follows:

**Corollary 2** (Multi-Time-Model of Intent- Planning Loss.). *Let $\mathcal{M}$ be any SMDP, $\mathcal{I}^\rightarrow$ a set of temporally extended multi-time-model of intents, $\mathcal{O}$ a set of options, and $\hat{M}_{\mathcal{AF}_{\mathcal{I}^\rightarrow}}$ the corresponding approximate SMDP over affordable state-option pairs $\mathcal{AF}_{\mathcal{I}^\rightarrow}$. Then, the certainty equivalence planning loss with $\hat{M}_{\mathcal{AF}_{\mathcal{I}^\rightarrow}}$ is*

$$\left\|V_{\mathcal{M}}^* - V_{\mathcal{M}}^{\pi^*_{\hat{\mathcal{M}}_{\mathcal{AF}_{\mathcal{I}^\rightarrow}}}}\right\|_\infty \leq \frac{2R_{max}^{\mathcal{O}}}{(1-\gamma^{\mathcal{O}})^2}\left(2\gamma\zeta^{\mathcal{I}^\rightarrow} + \sqrt{\frac{1}{2n}\log\frac{2|\mathcal{AF}_{\mathcal{I}^\rightarrow}||\Pi_{\mathcal{I}^\rightarrow}|}{\delta}}\right)$$

*with probability at least $1-\delta$, where $\zeta^{\mathcal{I}^\rightarrow}$ is the degree of satisfaction of the intents (Eq. 1), $R_{max}^{\mathcal{O}} = \max_{s,o} r(s,o)$ is the maximum option reward, and $\gamma^{\mathcal{O}} = \max_{s,o}\sum_{s'}\gamma_o(s,s')$ is the maximum expected discount factor for both intent and option model.*

*Proof.* We now show that the trajectories-based planning loss bound can be reduced to the special case where intents were defined via sub-probability distributions incorporating both time and final state.

First, we consider the trajectories-based planning loss bound:

$$\left\|V_{\mathcal{M}}^* - V_{\mathcal{M}}^{\pi^*_{\hat{\mathcal{M}}_{\mathcal{AF}_{\mathcal{I}}}}}\right\|_\infty \leq \frac{5\zeta_R^{\mathcal{I}}}{\left(1-\gamma^{\mathcal{I}}\right)} + \frac{2R_{max}^{\mathcal{O}}}{\left(1-\gamma^{\mathcal{I}}\right)\left(1-\gamma^{\mathcal{O}}\right)}\left(2\max_{s,o}\sum_{t=1}^{\infty}\gamma^t|\mathcal{S}|\zeta_P^{\mathcal{I}} + \sqrt{\frac{1}{2n}\log\frac{2|\mathcal{AF}_{\mathcal{I}}||\Pi_{\mathcal{I}}|}{\delta}}\right)$$

We plug our assumption that rewards are known and given which results in the constant $\zeta_R^{\mathcal{I}} = 0$, option and intent discount factors are assumed to be the same i.e. $\gamma^{\mathcal{O}} = \gamma^{\mathcal{I}}$, and the second term can be simplified further as follows:

$$\left\|V_{\mathcal{M}}^* - V_{\mathcal{M}}^{\pi^*_{\hat{\mathcal{M}}_{\mathcal{AF}_{\mathcal{I}}}}}\right\|_\infty \le \frac{2R_{max}^{\mathcal{O}}}{\left(1-\gamma^{\mathcal{O}}\right)^2} \times \Big(2\underbrace{\max_{s,o} \sum_{\tau(s,t,s')} \gamma(\tau(s,t,s'))\Big|P(\tau(s,t,s')|o) - P_I(\tau(s,t,s')|o)\Big|}_{\le \gamma\zeta^{\mathcal{I}}} +$$

$$\tag{12}$$

$$\sqrt{\frac{1}{2n} \log \frac{2|\mathcal{AF}_{\mathcal{I}}||\Pi_{\mathcal{I}}|}{\delta}}\Big)$$

$$\le \frac{2R_{max}^{\mathcal{O}}}{\left(1-\gamma^{\mathcal{O}}\right)^2} \times \Big(2\gamma\zeta^{\mathcal{I}} + \sqrt{\frac{1}{2n} \log \frac{2|\mathcal{AF}_{\mathcal{I}}||\Pi_{\mathcal{I}}|}{\delta}}\Big) \tag{13}$$

$\square$

### A.4 Intent expression on end-state

Consider the definition of $Q^*(s,o)$ from Sec. 3 and note that it can be re-written in our notation as:

$$Q^*(s,o) = \sum_{s'}(r(s,o,s') + \gamma_o(s,s')\max_{o'} Q^*(s',o'))$$

Note that $\gamma_o(s,s') \le \gamma$. With this notation, it is clear that the previous results from Sec. A.2 and Sec. A.3 on value loss and planning loss from Khetarpal et al. (2020a) apply readily. In particular, if options only take a single step, we recover exactly their bounds, as the reward difference upper bound $\zeta_R^{\mathcal{I}}$ will be 0 and the above inequality becomes equality i.e. $\gamma_o(s,s') = \gamma$.

## B  Details of Experiments

### B.0.1  Implementation Details

We use the environment implementation from OpenAI Gym[5]. We build upon open source code released by Khetarpal et al. (2020a) significantly scaling it up using Launchpad (Yang et al., 2021). Our code can be found at https://github.com/deepmind/affordances_option_models/. We implemented three nodes:

1. Data collection (Rollout): Runs options, $\pi_o(a|s)$, in the environment to collect transition data.
2. Model (and affordance) learning (Trainer): Uses the data from the Rollout node to train the option models and affordance models where relevant.
3. Planning and evaluation (Evaluation): Uses the trained options models to perform value iteration and obtain a policy over options, $\pi_{\mathcal{O}}(o_t|s_t)$. The policy over options, $\pi_{\mathcal{O}}(o_t|s_t)$, and options, $\pi_o(a|s)$, are then evaluated over 1000 episodes to record the proportion that successfully dropped the passenger.

We used a shared internal cluster and each run used $\approx 3$ cpus for $\approx 48$ hours. We used linear networks for all models. We initialize the affordance classifier to output 1 by shifting the input to the final sigmoid by 2, i.e. $A_\theta(s,o,s',I) = sigmoid(f_\theta(s,o,s',I) + 2)$, where $f_\theta$ is a linear model.

### B.0.2  Hyperparameter Settings

Given the simplicity and purpose of our experiments we only did a hyperparameter sweep over the learning rate (0.001, 0.0001). We chose the maximum option length to be a 100 to allow options to terminate naturally. We set the hidden size of the models to be 0 (i.e. linear models). Each experiment was repeated for 4 independent seeds. We use the color-blind friendly palette from Lawlor (2020) for our figures.

---

[5]https://github.com/openai/gym/blob/master/gym/envs/toy_text/taxi.py

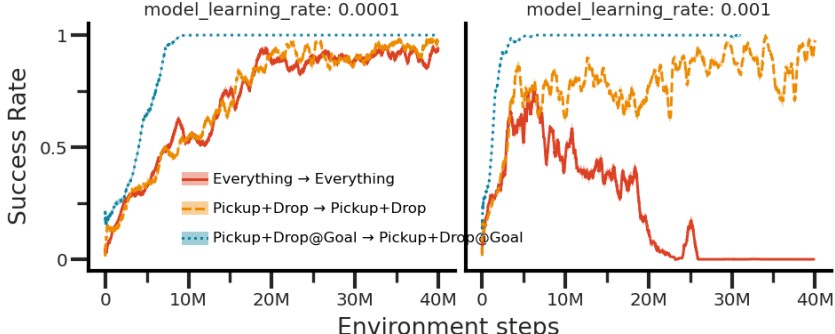

Figure B1: **A higher learning rate can be used to learn the model when using affordances**. Right shows divergence when using an unrestricted affordance set (Everything) for a higher learning rate compared to using any affordances.

## C  Sample Complexity Analysis - Multi-Time-Model of Intent

Classical methods for planning in RL assume access to the complete knowledge of the MDP. However, in large domains, this is an infeasible assumption. A common approach is to consider sample-based models in which the transitions are estimated by sampling the model, with the number of calls to this sampler referred to as the sample complexity. In practise, a model $\hat{P}$ is estimated to approximate the transition model which is then used for planning (See Sec 4.2).

We then ask the question of how difficult is to build an approximate model for everything in an environment. It is intuitive to see that modelling one-time step dynamics would require samples in the order of magnitude of the size of the state-action space (See Table 3). To mitigate this, we propose constructing temporally abstract partial models. Specifically, we examine the sample complexity of obtaining an $\varepsilon$ estimation of the optimal action-value function given only access to a generative model (Kearns and Singh, 1999; Kakade et al., 2003; Azar et al., 2012).

Consider a SMDP $\mathcal{M}$ where a deterministic policy over options is a map $\pi : \mathcal{S} \to \mathcal{O}$ that maps a state into an option. The value function of a policy $\pi$ is a vector $V^\pi \in \mathbb{R}^{|\mathcal{S}|}$, defined as follows, $\forall s \in \mathcal{S}$:

$$V^\pi(s) := \sum_{o \in \mathcal{O}} \pi(o|s) \left[ r(s, o) + \sum_{s'} p(s'|s, o) V^\pi(s') \right],$$

where $p(s'|s, o) = \sum_{k=1}^{\infty} p(s', k)\gamma^k$.

Analogously, the option value function $Q^\pi \in \mathbb{R}^{|\mathcal{S} \times \mathcal{O}|}$, for a policy $\pi$ is defined as follows, $\forall s \in \mathcal{S} \times \mathcal{O}$

$$Q^\pi(s, o) := r(s, o) + (P_o \cdot V^\pi)(s, o),$$

where

$$P_o = \sum_{s'} p(s'|s, o), \quad V^\pi(s') = \sum_{o' \in \mathcal{O}'} \pi(o'|s') Q^\pi(s', o')$$

As described earlier, we assume access to a generative model, which can provide us with samples at the SMDP level $\{s', \tau\} \sim P(\cdot|s, o)$. Similar to previously described setting, we consider a set of *temporally extended intents* $\mathcal{I}^\to$, with the assumption that each option $o$ has an intent associated with it $I_o$, resulting in an induced SMDP $\mathcal{M}_\mathcal{I}$, with corresponding option models denoted by $P_o^I$. Let $\hat{\mathcal{M}}_{\mathcal{AF}_{\mathcal{I}\to}}$ be the approximate SMDP over affordable state-option pairs denoted by $\mathcal{AF}_{\mathcal{I}\to}$, with $\hat{P}_o^I$ as the corresponding options model.

We then define $\hat{P}_o^I$, our empirical model for each option $o \in \mathcal{O}$ be defined as follows. $\forall o \in \mathcal{O}$:

$$\hat{P}_I(s'|s, o) = \frac{\texttt{count}(s, o, s')}{N} = \frac{\sum_{i=1}^{N} 1\{s_i' = s'\}\gamma^{\tau_i}}{N},$$
$$\texttt{where } \{s_i', \tau_i\} \sim P(\cdot|s, o) \forall 1 \le i \le N.$$

| | Sample Complexity | |
| --- | --- | --- |
| **Actions** | **Without Affordances** | **Affordance-aware** |
| **Primitive** | $\mathcal{O}\left(\frac{\|\mathcal{S}\|\|\mathcal{A}\|}{(1-\gamma)^4\varepsilon^2}\right)$ | $\mathcal{O}\left(\frac{\|\mathcal{AF}_{\mathcal{I}}\|}{(1-\gamma)^4\varepsilon^2}\right)$ |
| **Temporally Extended** | $\mathcal{O}\left(\frac{\|\mathcal{S}\|\|\mathcal{O}\|}{(1-\gamma)^4\varepsilon^2}\right)$ | $\mathcal{O}\left(\frac{\|\mathcal{AF}_{\mathcal{I}}\|}{(1-\gamma)^4\varepsilon^2}\right)$ |

Table 3: **Comparison of Sample Complexity** - provides evidence on the role of temporal abstraction and affordances in obtaining an $\varepsilon$ estimation of the optimal value function. Incorporating affordances results in potential improvements in sample complexity in both primitive and temporally extended actions, although at the cost of approximation error induced via intents. Here $\gamma$ is the maximum expected discount factor for both intent and option model.

where `count` is the number of times the state-option pair $(s, o)$ pair transitions to $s'$. Note that $\mathcal{M}_{\mathcal{I}}$ and $\hat{\mathcal{M}}_{\mathcal{AF}_{\mathcal{I}\rightarrow}}$ are equivalent to the SMDP $\mathcal{M}$ in reward[6], except the estimated transition dynamics instead of the true transition kernel per option i.e. $P_o$.

To derive an $\varepsilon$ optimal estimate of the optimal value function in the SMDP, we here consider the *SMDP Q-value iteration (QVI)* (Sutton et al., 1999) analogous to the primitive case of Q-value iteration, but only for state-option pairs that are affordable. See C.1.1 for details.

**Theorem 3.** *Let $\mathcal{M}$ be a SMDP, $\mathcal{I}^{\rightarrow}$ a set of temporally extended intents corresponding to a set of options $\mathcal{O}$. If $\hat{\mathcal{M}}_{\mathcal{AF}_{\mathcal{I}\rightarrow}}$ is the corresponding approximate SMDP over affordable state-option pairs $\mathcal{AF}_{\mathcal{I}\rightarrow}$, and $Q_k$ is returned by SMDP Q-value iteration at the $k^{th}$ epoch, with inputs including the approximate SMDP as the generative model, and number of samples $m$, where*

$$m = \mathcal{O}\left(\frac{|\mathcal{AF}_{\mathcal{I}\rightarrow}|}{(1-\gamma)^4\varepsilon^2}\right),$$

*then with probability greater than $1 - \delta$, the following holds for $\varepsilon \geq \frac{2\zeta^{\mathcal{I}^{\rightarrow}}\gamma}{(1-\gamma)^2}$, and for all $s, o$:*

$$||Q_k - Q^*||_\infty \leq \varepsilon,$$

*where $\zeta^{\mathcal{I}^{\rightarrow}}$ is the degree of satisfaction of the intents, $\gamma$ is the maximum expected discount factor of an option, $k = \frac{\log\left(\frac{\varepsilon(1-\gamma)^2 - 2\zeta^{\mathcal{I}^{\rightarrow}}\gamma}{2(1-\gamma)}\right)}{\log\gamma}$, and $Q^*$ is the optimal option value function in the underlying SMDP $\mathcal{M}$.*

The proof is in Appendix C.1.2. The approximation error in the intended distribution $\zeta^{\mathcal{I}^{\rightarrow}}$ predominantly governs how good an estimate of the optimal option value function can be made for a given set of intents $\mathcal{I}^{\rightarrow}$. Our results suggests that we can only guarantee approximations of $Q^*$ up to the lower bound on $\varepsilon$ i.e. $\frac{2\zeta^{\mathcal{I}^{\rightarrow}}\gamma}{(1-\gamma)^2}$.

Following through the proof of Theorem 3, it is easy to show that the number of samples $m$ required to obtain an $\varepsilon$ estimation of the optimal $Q$-value function without incorporating affordances is proportional to the size of the state-option space as shown in Theorem 4.

**Theorem 4.** *Let $\mathcal{M}$ be a SMDP with a set of options $\mathcal{O}$. If $\hat{\mathcal{M}}$ is the corresponding approximate SMDP, and $Q_k$ is returned by SMDP Q-value iteration at the $k^{th}$ epoch, with inputs including the approximate SMDP as the generative model, and number of samples $m$, where*

$$m = \mathcal{O}\left(\frac{|\mathcal{S}||\mathcal{O}|}{(1-\gamma)^4\varepsilon^2}\right),$$

*then with probability greater than $1 - \delta$, the following holds for all $s$ and $o$:*

$$||Q_k - Q^*||_\infty \leq \varepsilon,$$

*where $\gamma$ is the maximum expected option discount factor, $k = \frac{\log(\varepsilon(1-\gamma))}{\log\gamma}$, and $Q^*$ is the optimal option value function in the underlying SMDP $\mathcal{M}$.*

---

[6]Note that here we assume the reward function is known and deterministic and therefore is identical to the true SMDP.

For a complete proof, See Appendix C.1.3. To summarize, Table 3 decouples the role of temporal abstraction and the effect of incorporating affordances. Predicting and reasoning across multiple timescales naturally results in a growing set of action choices leading to a large number of samples. Larger gains can be established when considering both temporal abstractions and affordance information, with a carefully designed set of intents.

## C.1 Proofs - Sample Complexity Analysis

**Note:** We again overload notation and throughout our proofs, for convenience we interchangeably use $\mathcal{I}$ and $\mathcal{I}^{\rightarrow}$ to denote set of temporally extended intents. Similarly, for convenience we interchangeably use $I$ and $\mathrm{I}_o^{\rightarrow}$ to denote a temporally extended intent for an option $o$.

### C.1.1 SMDP Q-Value Iteration (QVI)

To derive an $\varepsilon$ optimal estimate of the optimal option-value function in the SMDP, we here consider the *SMDP Q-value iteration* (SMDP-QVI) (Sutton et al., 1999) process as detailed in algorithm below.

---

**Algorithm 1 Model-based SMDP Q-Value Iteration (SMDP-QVI)**

1: $V_0 = 0, Q_0 = 0$
2: **for** epoch $k = 1 \ldots K$ **do**
3:    **for** $(s, o) \in \mathcal{AF}_\mathcal{I}$, **do**
4:       $Q_k(s, o) = r(s, o) + (\hat{P}_o^I V_{k-1})(s, o)$
5:       $V_k(s) = \max_{o \in \mathcal{AF}_\mathcal{I}(s)} Q_k(s, o)$
6:    **end for**
7: **end for**
8: Output $Q_k$

---

### C.1.2 Proof of Theorem 3 - Sample complexity of Temporally Abstract Partial Model.

*Proof.* We here consider the transition models in the ground SMDP $\mathcal{M}$, the intent induced SMDP $\mathcal{M}_\mathcal{I}$, and the approximate SMDP $\hat{\mathcal{M}}_{\mathcal{AF}_\mathcal{I}}$ over affordable state-option pairs are denoted by $P_o$, $P_o^I$, and $\hat{P}_o^I$ respectively.

We here consider $\left\|Q_k - Q^*\right\|_\infty$.

Adding and subtracting $\hat{Q}^*_{\hat{\mathcal{M}}_{\mathcal{AF}_\mathcal{I}}}$ and $Q^*_{\mathcal{M}_\mathcal{I}}$ we get,

$$Q_k - Q^* = \underbrace{Q_k - \hat{Q}^*_{\hat{\mathcal{M}}_{\mathcal{AF}_\mathcal{I}}}}_{\text{Term (A)}} + \underbrace{\hat{Q}^*_{\hat{\mathcal{M}}_{\mathcal{AF}_\mathcal{I}}} - Q^*_{\mathcal{M}_\mathcal{I}}}_{\text{Term (B)}} + \underbrace{Q^*_{\mathcal{M}_\mathcal{I}} - Q^*}_{\text{Term (C)}}$$

**Bounding Term (A)**

$$\left\|Q_k - \hat{Q}^*_{\hat{\mathcal{M}}_{\mathcal{AF}_\mathcal{I}}}\right\|_\infty = \max_{(s,o)\in\mathcal{AF}_\mathcal{I}} \left[ r(s,o) + (\hat{P}_o^I V_{k-1})(s,o) - (r(s,o) + (\hat{P}_o^I \hat{V}^*)(s,o)) \right]$$

$$= \max_{(s,o)\in\mathcal{AF}_\mathcal{I}} \left| (\hat{P}_o^I (V_{k-1} - \hat{V}^*))(s,o) \right|$$

$$\leq \gamma \left\| V_{k-1} - \hat{V}^* \right\|_\infty$$

$$\leq \gamma \max_{s\in\mathcal{AF}_\mathcal{I}(o)} \left| \max_{o\in\mathcal{AF}_\mathcal{I}(s)} Q_{k-1}(s,o) - \max_{o\in\mathcal{AF}_\mathcal{I}(s)} \hat{Q}^*_{\hat{\mathcal{M}}_{\mathcal{AF}_\mathcal{I}}}(s,o) \right|$$

$$\leq \gamma \max_{(s,o)\in\mathcal{AF}_\mathcal{I}} \left| Q_{k-1}(s,o) - \hat{Q}^*_{\hat{\mathcal{M}}_{\mathcal{AF}_\mathcal{I}}}(s,o) \right|$$

$$= \gamma \left\| Q_{k-1} - \hat{Q}^*_{\hat{\mathcal{M}}_{\mathcal{AF}_\mathcal{I}}} \right\|_\infty$$

Unrolling the above $k$ times, we get;

$$\left\|Q_k - \hat{Q}^*_{\hat{\mathcal{M}}_{\mathcal{AF_I}}}\right\|_\infty \le (\gamma)^k \left\|Q_0 - \hat{Q}^*_{\hat{\mathcal{M}}_{\mathcal{AF_I}}}\right\|_\infty \le \frac{(\gamma)^k}{(1-\gamma)}$$

**Bounding Term (B)**

$$
\begin{aligned}
\left(\hat{Q}^*_{\hat{\mathcal{M}}_{\mathcal{AF_I}}} - Q^*_{\mathcal{M_I}}\right)(s,o) &= (\hat{P}^I_o \hat{V}^*)(s,o) - (P^I_o V^*)(s,o) \\
&= \underbrace{(\hat{P}^I_o V^* - P^I_o V^*)(s,o)}_{} + \underbrace{(\hat{P}^I_o \hat{V}^*)(s,o) - (\hat{P}^I_o V^*)(s,o)}_{} \quad \text{Adding and Subtracting } \hat{P}^I_o V^* \\
&= \left(\left(\hat{P}^I_o - P^I_o\right)V^*\right)(s,o) - \left(\hat{P}^I_o\left(V^* - \hat{V}^*\right)\right)(s,o) \\
&= \left(\left(\hat{P}^I_o - P^I_o\right)V^*\right)(s,o) - \\
&\quad \sum_{s' \in \mathcal{AF_I}(o)} \hat{P}^I_o(s'|s,o)\left(\max_{o' \in \mathcal{AF_I}(s)} Q^*_{\mathcal{M_I}}(s',o') - \max_{o' \in \mathcal{AF_I}(s)} \hat{Q}^*_{\hat{\mathcal{M}}_{\mathcal{AF_I}}}(s',o')\right)
\end{aligned}
$$

Considering the max over all state-options, we have;

$$\left\|\hat{Q}^*_{\hat{\mathcal{M}}_{\mathcal{AF_I}}} - Q^*_{\mathcal{M_I}}\right\|_\infty \le \left\|\left(\hat{P}^I_o - P^I_o\right)V^*\right\| + \gamma \left\|\hat{Q}^*_{\hat{\mathcal{M}}_{\mathcal{AF_I}}} - Q^*_{\mathcal{M_I}}\right\|_\infty$$

Finally;

$$\left\|\hat{Q}^*_{\hat{\mathcal{M}}_{\mathcal{AF_I}}} - Q^*_{\mathcal{M_I}}\right\|_\infty \le \frac{1}{(1-\gamma)}\left\|\left(\hat{P}^I_o - P^I_o\right)V^*_{\mathcal{M_I}}\right\|$$

Now let's fix a state option pair $(s,o) \in \mathcal{AF_I}$

$$
\begin{aligned}
\left(\hat{P}^I_o - P^I_o\right)V^*_{\mathcal{M_I}} &= \frac{1}{N}\sum_{i=1}^N V^*_{\mathcal{M_I}}(s'_i) - \mathrm{E}_{s' \in P^I_o(\cdot|s,o)}[V^*_{\mathcal{M_I}}(s')] \\
&= \frac{1}{N}\left(S_N - \mathrm{E}[S_N]\right)
\end{aligned}
$$

where $S_N = \sum_{i=1}^N X_i$ and $X_i = V^*(s'_i)$, $X_i$ are independent variable and $|X_i| \le V_{max}$.
We now consider the Hoeffdings inequality:

$$P\left(\frac{1}{N}(S_N - \mathrm{E}[S_N]) \ge t\right) \le 2\exp\left(\frac{-N^2 t^2}{N V^2_{max}}\right) = 2\exp\left(\frac{-N t^2}{V^2_{max}}\right)$$

Applying Hoeffdings, we get;

$$
\begin{aligned}
P\left(\max_{s,o \in \mathcal{AF_I}}\left|(\hat{P}^I_o - P^I_o)V^*_{\mathcal{M_I}}(s,o)\right| \ge t\right) &= P\left(\exists(s,o \in \mathcal{AF_I})s.t.\left|(\hat{P}^I_o - P^I_o)V^*_{\mathcal{M_I}}(s,o)\right| \ge t\right) \\
&\le \sum_{\mathcal{AF_I}} Pr\left(\left|(\hat{P}^I_o - P^I_o)V^*_{\mathcal{M_I}}(s,o)\right| \ge t\right) \text{// Union Bound} \\
&= 2|\mathcal{AF_I}(o)||\mathcal{AF_I}(s)|\exp\left(\frac{-N t^2}{V^2_{max}}\right) \\
&= 2|\mathcal{AF_I}|\exp\left(\frac{-N t^2}{V^2_{max}}\right)
\end{aligned}
$$

We assume that the failure probability $\delta \geq 0$, We then solve for $t$ by equating the RHS to $\delta$ as follows:

$$2|\mathcal{AF}_{\mathcal{I}}| \exp\left(\frac{-Nt^2}{V_{max}^2}\right) = \delta$$

$$\exp\left(\frac{-Nt^2}{V_{max}^2}\right) = \frac{\delta}{2|\mathcal{AF}_{\mathcal{I}}|}$$

$$\frac{-Nt^2}{V_{max}^2} = \log\frac{\delta}{2|\mathcal{AF}_{\mathcal{I}}|}$$

$$t^2 = \frac{V_{max}^2}{N}\log\frac{2|\mathcal{AF}_{\mathcal{I}}|}{\delta}$$

$$t = V_{max}\sqrt{\frac{1}{N}\log\frac{2|\mathcal{AF}_{\mathcal{I}}|}{\delta}}$$

Plugging this back in Term (B) $\left\|\hat{Q}^*_{\hat{\mathcal{M}}_{\mathcal{AF}_{\mathcal{I}}}} - Q^*_{\mathcal{M}_{\mathcal{I}}}\right\|_\infty \leq \frac{1}{(1-\gamma)}\left\|\left(\hat{P}_o - P_o\right)V^*\right\|$, we get:

$$\left\|\hat{Q}^*_{\hat{\mathcal{M}}_{\mathcal{AF}_{\mathcal{I}}}} - Q^*_{\mathcal{M}_{\mathcal{I}}}\right\|_\infty \leq \frac{V_{max}}{(1-\gamma)}\sqrt{\frac{1}{N}\log\frac{2|\mathcal{AF}_{\mathcal{I}}|}{\delta}}$$

Based on Remark 2,

$$\left\|\hat{Q}^*_{\hat{\mathcal{M}}_{\mathcal{AF}_{\mathcal{I}}}} - Q^*_{\mathcal{M}_{\mathcal{I}}}\right\|_\infty \leq \frac{R^{\mathcal{O}}_{max}}{(1-\gamma)^2}\sqrt{\frac{1}{N}\log\frac{2|\mathcal{AF}_{\mathcal{I}}|}{\delta}}$$

**Bounding Term (C)** $\left\|Q^*_{\mathcal{M}_{\mathcal{I}}} - Q^*\right\|_\infty$

We first define the following optimality bellman operator:

$$Q^*_{\mathcal{M}} = \mathcal{T}Q^*_{\mathcal{M}}$$

$$\text{where}\left(\mathcal{T}f\right) := R(s,o) + \langle P(s,o), V_f\rangle$$

$$\text{where}V_f(\cdot) := \max_{o\in\mathcal{O}} f(\cdot, o)$$

Our aim here is to bound $\left\|Q^*_{M_1} - Q^*_{M_2}\right\|_\infty$ for any two SMDP models $M_1$ and $M_2$.

Let $\mathcal{T}_1$ and $\mathcal{T}_2$ be the Bellman operator of the SMDPs $M_1$ and $M_2$ respectively. Therefore,

$$\left\|Q^*_{M_1} - \mathcal{T}_2 Q^*_{M_1}\right\|_\infty = \left\|\mathcal{T}_1 Q^*_{M_1} - \mathcal{T}_2 Q^*_{M_1}\right\|_\infty$$

$$= \max_{(s,o)\in S\times\mathcal{O}}\left|\langle P_1(s,o), V^*_{M_1}\rangle - \langle P_2(s,o), V^*_{M_1}\rangle\right|$$

$$= \max_{(s,o)\in S\times\mathcal{O}}\left|\mathbb{E}_{s'\sim P_1(s,o)}[V^*_{M_1}(s')] - \mathbb{E}_{s'\sim P_2(s,o)}[V^*_{M_1}(s')]\right|$$

$$\leq \left\|d^{\mathrm{F}}_{M_1, M_2}\right\|_\infty$$

Therefore,

$$\left\|Q^*_{M_1} - Q^*_{M_2}\right\|_\infty = \left\|Q^*_{M_1} - \mathcal{T}_2 Q^*_{M_1} + \mathcal{T}_2 Q^*_{M_1} - \mathcal{T}_2 Q^*_{M_2}\right\|_\infty$$

$$\leq \left\|d^{\mathrm{F}}_{M_1, M_2}\right\|_\infty + \left\|\mathcal{T}_2 Q^*_{M_1} - \mathcal{T}_2 Q^*_{M_2}\right\|_\infty$$

Bounding the second term of the last step i.e. $\left\lVert \mathcal{T}_2 Q^*_{M_1} - \mathcal{T}_2 Q^*_{M_2} \right\rVert_\infty$ ;

$$\left| \mathcal{T}_2 f_1(s,o) - \mathcal{T}_2 f_2(s,o) \right| = \left| \big( r(s,o) + \langle P_2(s,o) V_{f_1}(s) \rangle \big) - \big( r(s,o) + \langle P_2(s,o) V_{f_2}(s) \rangle \big) \right|$$

$$= \left| \langle P_2(s,o) V_{f_1}(s) \rangle \rangle - \langle P_2(s,o) V_{f_2}(s) \rangle \rangle \right|$$

$$\leq \max_{(s,o) \in \mathcal{S} \times \mathcal{O}} \left| \mathbb{E}_{s' \sim P_2(s,o)}[V_{f_1}(s')] - \mathbb{E}_{s' \sim P_2(s,o)}[V_{f_2}(s')] \right|$$

$$= \max_{(s,o) \in \mathcal{S} \times \mathcal{O}} \sum_{s'} P_2(s'|s,o) \left| V_{f_1}(s') - V_{f_2}(s') \right|$$

$$\leq \gamma \left\lVert V^*_{M_1} - V^*_{M_2} \right\rVert_\infty$$

Therefore,

$$\left\lVert Q^*_{\mathcal{M}_\mathcal{I}} - Q^* \right\rVert_\infty \leq \left\lVert d^{\mathrm{F}}_{M_1, M_2} \right\rVert_\infty + \gamma \left\lVert V^*_{\mathcal{M}_\mathcal{I}} - V^* \right\rVert_\infty$$

where note that the second term in the last step is bounded as following,

$$\max_s \left| V^*_{M_1} - V^*_{M_2} \right| = \max_s \left| \max_o Q^*_{M_1}(s,o) - \max_o Q^*_{M_2}(s,o) \right|$$

$$\leq \max_s \left| \max_o (Q^*_{M_1}(s,o) - Q^*_{M_2}(s,o)) \right|$$

$$\leq \max_{s,o} \left| Q^*_{M_1}(s,o) - Q^*_{M_2}(s,o) \right|$$

$$= \left\lVert Q^*_{M_1} - Q^*_{M_2} \right\rVert_\infty$$

Therefore,

$$\left\lVert Q^*_{\mathcal{M}_\mathcal{I}} - Q^* \right\rVert_\infty \leq \left\lVert d^{\mathrm{F}}_{M_1, M_2} \right\rVert_\infty + \gamma \left\lVert V^*_{\mathcal{M}_\mathcal{I}} - V^* \right\rVert_\infty$$

$$\leq \left\lVert d^{\mathrm{F}}_{M_1, M_2} \right\rVert_\infty + \gamma \left\lVert Q^*_{\mathcal{M}_\mathcal{I}} - Q^* \right\rVert_\infty$$

$$\leq \frac{1}{(1-\gamma)} \left\lVert d^{\mathrm{F}}_{\mathcal{M}_\mathcal{I}, \mathcal{M}} \right\rVert_\infty$$

$$\leq \frac{\zeta^\mathcal{I} \gamma R^\mathcal{O}_{max}}{(1-\gamma)^2}.$$

We conclude,

$$\left\lVert Q_k - Q^* \right\rVert_\infty \leq \left\lVert Q_k - \hat{Q}^*_{\hat{\mathcal{M}}_{\mathcal{AF}_{\mathcal{I}\to}}} \right\rVert_\infty + \left\lVert \hat{Q}^*_{\hat{\mathcal{M}}_{\mathcal{AF}_{\mathcal{I}\to}}} - Q^*_{\mathcal{M}_\mathcal{I}} \right\rVert_\infty + \left\lVert Q^*_{\mathcal{M}_\mathcal{I}} - Q^* \right\rVert_\infty$$

$$\leq \frac{(\gamma)^k}{(1-\gamma)} + \frac{1}{(1-\gamma)^2} \sqrt{\frac{1}{N} \log(2|\mathcal{AF}_\mathcal{I}|)} + \frac{\zeta^\mathcal{I} \gamma R^\mathcal{O}_{max}}{(1-\gamma)^2}$$

To obtain an $\varepsilon$ estimation of the optimal $Q$-value function in the SMDP, we distribute the error across Term A, B, and C such that ;

$$\left\lVert Q_k - Q^* \right\rVert_\infty \leq \underbrace{\text{Term (A)} + \text{Term (C)}}_{\leq \varepsilon/2} + \underbrace{\text{Term (B)}}_{\leq \varepsilon/2}$$

By choosing $k = \dfrac{\log\left( \frac{\varepsilon(1-\gamma)^2 - 2\zeta\gamma}{2(1-\gamma)} \right)}{\log \gamma}$ and $N = \frac{4}{(1-\gamma)^4 \varepsilon^2} \log(2|\mathcal{AF}_{\mathcal{I}\to}|)$, we get $\left\lVert Q_k - Q^* \right\rVert_\infty \leq \varepsilon/2 + \varepsilon/2$

Note that this choice of $k$ holds if and only if:

$$\varepsilon(1-\gamma)^2 \geq 2\zeta^{\mathcal{I}}\gamma$$

$$\varepsilon \geq \frac{2\zeta^{\mathcal{I}}\gamma}{(1-\gamma)^2}$$

Therefore, the total number of samples needed to get an $\varepsilon$-estimation of the optimal option value function is;

$$N|\mathcal{S}||\mathcal{O}| = \mathcal{O}\Big(\frac{|\mathcal{AF}_{\mathcal{I}\to}|}{(1-\gamma)^4\varepsilon^2}\Big)$$

$\square$

### C.1.3 Proof of Theorem 4 - Sample complexity of Temporally Abstract Full Model.

*Proof.* We here consider $\left\|Q_k - Q^*\right\|_\infty$, and $Q^*$ is the optimal option value function in the underlying SMDP $\mathcal{M}$.

Adding and subtracting $\hat{Q}^*$ we get,

$$Q_k - Q* = \underbrace{Q_k - \hat{Q}^*}_{\text{Term (A)}} + \underbrace{\hat{Q}^* - Q^*}_{\text{Term (B)}}$$

**Bounding Term (A)**

$$\begin{aligned}
\left\|Q_k - \hat{Q}^*\right\|_\infty &= \max_{(s,o)\in\mathcal{S}\times\mathcal{O}} \Big[r(s,o) + \hat{P}_o V_{k-1}(s,o) - (r(s,o) + \hat{P}_o\hat{V}^*(s,o))\Big] \\
&= \max_{(s,o)\in\mathcal{S}\times\mathcal{O}} \Big|\hat{P}_o(V_{k-1} - \hat{V}^*)(s,o)\Big| \\
&\leq \gamma^{\mathcal{D}}\left\|V_{k-1} - \hat{V}^*\right\|_\infty \\
&\leq \gamma^{\mathcal{D}}\max_{s\in\mathcal{S}}\Big|\max_{o\in\mathcal{O}}Q_{k-1}(s,o) - \max_{o\in\mathcal{O}}\hat{Q}^*(s,o)\Big| \\
&\leq \gamma^{\mathcal{D}}\max_{(s,o)\in\mathcal{S}\times\mathcal{O}}\Big|Q_{k-1}(s,o) - \hat{Q}^*(s,o)\Big| \\
&= \gamma^{\mathcal{D}}\left\|Q_{k-1} - \hat{Q}^*\right\|_\infty
\end{aligned}$$

Unrolling the above $k$ times, we get;

$$\left\|Q_k - \hat{Q}^*\right\|_\infty \leq (\gamma^{\mathcal{D}})^k\left\|Q_0 - \hat{Q}^*\right\|_\infty \leq \frac{(\gamma^{\mathcal{D}})^k}{(1-\gamma^{\mathcal{D}})}$$

**Bounding Term (B)**

$$\begin{aligned}
\left(\hat{Q}^* - Q^*\right)(s,o) &= \hat{P}_o\hat{V}^*(s,o) - P_o V^*(s,o) \\
&= \hat{P}_o V^*(s,o) - P_o V^*(s,o) - \hat{P}_o\hat{V}^*(s,o) - \hat{P}_o V^*(s,o) \text{ Adding and Subtracting } \hat{P}_o V^* \\
&= \left(\hat{P}_o - P_o\right)V^*(s,o) - \hat{P}_o\left(\hat{V}^* - V^*\right)(s,o) \\
&= \left(\hat{P}_o - P_o\right)V^*(s,o) - \sum_{s'\in\mathcal{S}}\hat{P}_o(s'|s,o)\Big(\max_{o'\in\mathcal{O}}\hat{Q}^*(s',o') - \max_{o'\in\mathcal{O}}Q^*(s',o')\Big)
\end{aligned}$$

Therefore;

$$\left\|\hat{Q}^* - Q^*\right\|_\infty \leq \left\|\left(\hat{P}_o - P_o\right)V^*\right\| + \gamma^{\mathcal{D}}\left\|\hat{Q}^* - Q^*\right\|_\infty$$

Finally;

$$\left\|\hat{Q}^* - Q^*\right\|_\infty \le \frac{1}{(1-\gamma^{\mathcal{D}})}\left\|\left(\hat{P}_o - P_o\right)V^*\right\|$$

Now let's fix a state option pair $(s,o) \in \mathcal{S} \times \mathcal{O}$

$$\left(\hat{P}_o - P_o\right)V^* = \frac{1}{N}\sum_{i=1}^{N} V^*(s_i') - \mathrm{E}_{s' \in P_o(\cdot|s,o)}[V^*(s')]$$

$$= \frac{1}{N}\left(S_N - \mathrm{E}[S_N]\right)$$

where $S_N = \sum_{i=1}^{N} X_i$ and $X_i = V^*(s_i')$, $X_i$ are independent variable and $|X_i| \le V_{max}$.
We now consider the Hoeffdings inequality:

$$P\left(\frac{1}{N}(S_N - \mathrm{E}[S_N]) \ge t\right) \le 2\exp\left(\frac{-N^2 t^2}{N V_{max}^2}\right) = 2\exp\left(\frac{-N t^2}{V_{max}^2}\right)$$

Applying Hoeffdings, we get;

$$P\left(\max_{\mathcal{S},\mathcal{O}}\left|(\hat{P}_o - P_o)V^*(s,o)\right| \ge t\right) = P\left(\exists(s,o)s.t.\left|(\hat{P}_o - P_o)V^*(s,o)\right| \ge t\right)$$

$$\le \sum_{\mathcal{S},\mathcal{O}} Pr\left(\left|\left(\hat{P}_o - P_o\right)V^*(s,o)\right| \ge t\right) \text{// Union Bound}$$

$$= 2|\mathcal{S}||\mathcal{O}|\exp\left(\frac{-N t^2}{V_{max}^2}\right)$$

We assume that the failure probability $\delta \ge 0$, We then solve for $t$ by equating the RHS to $\delta$ as follows:

$$2|\mathcal{S}||\mathcal{O}|\exp\left(\frac{-N t^2}{V_{max}^2}\right) = \delta$$

$$\exp\left(\frac{-N t^2}{V_{max}^2}\right) = \frac{\delta}{2|\mathcal{S}||\mathcal{O}|}$$

$$\frac{-N t^2}{V_{max}^2} = \log\frac{\delta}{2|\mathcal{S}||\mathcal{O}|}$$

$$t^2 = \frac{V_{max}^2}{N}\log\frac{2|\mathcal{S}||\mathcal{O}|}{\delta}$$

$$t = V_{max}\sqrt{\frac{1}{N}\log\frac{2|\mathcal{S}||\mathcal{O}|}{\delta}}$$

Plugging this back in Term (B) $\left\|\hat{Q}^* - Q^*\right\|_\infty \le \frac{1}{(1-\gamma^{\mathcal{D}})}\left\|\left(\hat{P}_o - P_o\right)V^*\right\|$, we get:

$$\left\|\hat{Q}^* - Q^*\right\|_\infty \le \frac{V_{max}}{(1-\gamma^{\mathcal{D}})}\sqrt{\frac{1}{N}\log\frac{2|\mathcal{S}||\mathcal{O}|}{\delta}}$$

Therefore;

$$\left\|Q_k - Q^*\right\|_\infty \le \left\|Q_k - \hat{Q}^*\right\|_\infty + \left\|\hat{Q}^* - Q^*\right\|_\infty$$

$$\le \frac{(\gamma^{\mathcal{D}})^k}{(1-\gamma^{\mathcal{D}})} + \frac{V_{max}}{(1-\gamma^{\mathcal{D}})}\sqrt{\frac{1}{N}\log\frac{2|\mathcal{S}||\mathcal{O}|}{\delta}}$$

$$\le \frac{(\gamma^{\mathcal{D}})^k}{(1-\gamma^{\mathcal{D}})} + \frac{R_{max}}{(1-\gamma^{\mathcal{D}})^2}\sqrt{\frac{1}{N}\log\frac{2|\mathcal{S}||\mathcal{O}|}{\delta}}$$

To obtain an $\varepsilon$ estimation of the optimal $Q$-value function in the SMDP, we distribute the error uniformly;

$$\left\|Q_k - Q^*\right\|_\infty \leq \varepsilon/2 + \varepsilon/2$$

Equating each term to $\varepsilon/2$ and solving for $k$ and $N$ results in $k = \frac{\log(\varepsilon(1-\gamma^{\mathcal{D}}))}{\log \gamma^{\mathcal{D}}}$ and $N = \frac{4}{(1-\gamma^{\mathcal{D}})^4 \varepsilon^2} \log(2|\mathcal{S}||\mathcal{O}|)$ Therefore;

$$N|\mathcal{S}||\mathcal{O}| = \mathcal{O}\left(\frac{|\mathcal{S}||\mathcal{O}|}{(1 - \gamma^{\mathcal{D}})^4 \varepsilon^2}\right)$$

$\square$