# OpenReview forum: "Temporally Abstract Partial Models"
_NeurIPS.cc/2021/Conference — NeurIPS 2021 Poster_

### Official Review · Reviewer_VrS9 · 2021-07-02

**Rating:** 6
**Confidence:** 3

**Summary:**

This paper studies an option-based setting where each option is associated with an _intent_, a distribution over desired trajectories following the option execution. Intents induce _affordances_,  which are starting states that are likely to lead to the intents for each option. Intents and affordances induce a partial option model, where the usual transition model over options is replaced with the intents, and the model is partial over the affordances. The main theoretical questions concern the extent to which planning in this partial option model will lead to good performance in the original semi-MDP. Experiments are in a standard taxi domain with given options and intents.

**Limitations And Societal Impact:**

See “concerns” above.


**Main Review:**

Positive comments:
* Very impressive engagement with previous literature throughout.
* Given the previous work (Khetarpal et al., 2020a), this work is a well-motivated extension and a satisfying generalization of the previous results.
* I like that the generalization considers intents over full trajectories, rather than just terminal states; the latter would have been fine too, but it is good to remain as general as possible.
* The theoretical results are of some interest on their own, irrespective of applications.
* I did not find any technical errors, though I have some confusions (see below).
* The affordance learning results in Section 5.2 are clear and convincing.
* The paper is very well written on the sentence level, with a few minor typos noted below.


Concerns and criticisms:
* This question would broadly apply to the previous work (Khetarpal et al., 2020a) just as much, but I am wondering if this line of work is meant to be “descriptive” or “prescriptive”. Is the idea that we want to analyze an intelligent agent that comes prepackaged with possibly suboptimal options and intents, which may be poorly matched with each other (descriptive)? Or is this work suggesting that we should design and learn an agent architecture involving options and intents (prescriptive)? From a prescriptive perspective, I would struggle to see the benefit of introducing intents and affordances into the standard option framework. A descriptive perspective could be interesting, but then I wonder if the proposed models would really be used to model natural phenomena.
* The paper is difficult to read after the first two sections. This is partly because there is a lot of dense content, but there are also some opportunities to improve clarity without necessarily increasing word count. See also “confusions” below.
* One thing that would improve clarity would be to have a dedicated discussion reviewing (Khetarpal et al., 2020a). I ended up reading this previous work, which provided a lot of useful context.
* It would be useful to formally define the intent-induced MDP; right now it is just described informally in L184-185.
* While space is limited, I would have liked to see more discussion of the implications of, and general intuition behind, the main theoretical results, especially how they are interestingly different from the results obtained in the previous work.
* In the experiments, my understanding is that intents are always given, but the affordances are either given (section 5.1) or learned (section 5.2). However, the intents are only explicitly discussed in section 5.2. If my understanding is correct, it would be much more clear to define the intents along with the other environment details at the beginning of section 5.
* My reading of L276-278 is that (model learning + planning) with affordances is claimed to be better than (model learning alone), but Figure 3(c) vs 3(a) does not support that claim (the results look almost identical).
* The title could be much more specific.


Confusions:
* I am not sure what “sub-probability intent” means in Table 1
* It would have been helpful to include code in the supplementary materials
* In the experiments, it is not clear whether the state representation is tabular or something vector-based
* L273 - 274 mentions option misspecification, I am not sure what that is referring to
* It is sometimes not clear whether “affordance” or “affordable” is referring to both intents and affordances, or affordances alone
* “Model learning alone” in section 5.1 is a bit of a confusing name, because I suspect that almost all of the limitations of this baseline are due to the affordances being not used during data collection (as opposed to masking the loss). Something like “data collection” may be more clear.


Minor:
* L46 typo: “a partial, approximate models”
* L124 typo: “And intent”
* L125 typo: “drop of the passenger”
* L127 typo: “consequences of executing option.”
* The placement of Table 1 is a bit distracting, it’s unclear whether we are supposed to read it immediately after the text above
* In Equation 1, I don’t think that $\tau(s)$ was previously defined (I only see $\tau(t)$ and $\tau(s, s’)$), but I can understand from context that it represents the trajectory distribution starting at $s$ and executing the option.
* L199 typo: “corolaries”
* Table 2 Rmax has typesetting issues in the top row
* The notation for intention/affordance-induced MDP switches between including and excluding AF in the subscript



**Time Spent Reviewing:**

6

---

> ### Author Response · Authors · 2021-08-10
> **Response to Reviewer VrS9**
>
> Thank you very much for your thorough feedback and support of our paper submission. We address your comments below.
>
>
> **Descriptive vs Prescriptive**
>
> This is a great insight. We really appreciate your comments about this. Our work is meant to lean towards descriptive. Consider a never-ending scenario where the environment dynamics are unknown and might be non-stationary. An agent may need to  deal with lots of options and intents to manage different skills. Since the environment dynamics are non-stationary, affordances will allow the agent to maintain an efficient, focused, model that matters, irrespective of the number of options and intents.
>
> **Clarity and Presentation**
>
> Thank you for the detailed suggestions. We have incorporated these writing suggestions to enhance the paper further. We agree that a separate review of Khetarpal et. al (2020) will alleviate understanding our submission even better.
>
> **Confusions**
>
>  We have clarified all the confusions in our writing changes including 1. open sourcing code, 2. correction on model learning and data collection terminology, 3. clarification in writing on intent induced MDP, 3. clarification on state representation: There are a total of $25$ (grid positions) $\times 4$  (goal destinations) $\times 5$  (passenger scenarios) $= 500$ states in this environment and the observation is a one-hot vector among other typos and edits.
>
> **Fig 3c vs 3a**
>
> Regarding, L276-278 Fig 3c vs 3a which are identical and do not support the claim that (model learning + planning) with affordances is better than (model learning alone): This is a great catch we will fix this plot that was erroneously provided, however the statement still holds. We will update the figures and update the styling of the plots to make the difference apparent since the curves overlap in the later stages of learning.
>
> We believe that the paper and in particular its presentation have greatly benefited from the reviewers’ insightful feedback. We welcome any further comments or additional questions.

---

> > ### Comment · Reviewer_VrS9 · 2021-08-14
> > **Thanks; share new figure?**
> >
> > Thank you for addressing my comments and questions. I wonder if it would be possible to anonymously share a version of figure 3 with the update you mentioned.

---

> > > ### Author Response · Authors · 2021-08-16
> > > **Updated Figure 3**
> > >
> > > Thank you for your response. Please find an anonymous link to [Figure 3 here](https://upload.vaa.red/27qTAi#3ae71e35937339de99e6f24a37260270).
> > >
> > > Please let us know if you have any further comments or additional questions. We are happy to address them.

---

### Official Review · Reviewer_R8s1 · 2021-07-15

**Rating:** 6
**Confidence:** 3

**Summary:**

This paper addresses the problem of learning and planning with options framework in RL. The authors extend an existing notations of affordness and intent in the context of options and perform a mathematical analysis of the bound and the value loss. The ideas are demonstrated on a taxi benchmark with predefined set of options to illustrates the benfit of the approach.



**Limitations And Societal Impact:**

Yes

**Main Review:**

Most or even all of the ideas in the paper were already introduced and analyzed in previous: K. Khetarpal et. al. “What can I do here? A Theory of Affordness in Reinforcement Learning”, the main contribution here is the definition and extension of the framework to trajectories. The same bound and analysis was done in that light and appropriate bounds were derived. A missing piece in all this line of work of planning with value loss is the comparison to regret minimization in the context of options. At least something should be said about the differences between the two approaches. The paper in well structured and written clearly, and all of the bounds are aligned with previous results.

Minor issues
1. p.4 equation for probability under option o, there is a missing comma at the beginning <sa0,.. →< s,a0...

2. p. 7 “Experimental pipeline”, T is not defined properly. Is it a constant for all option trajectories (hyper-parameter)? an RV?

**Time Spent Reviewing:**

1

---

> ### Author Response · Authors · 2021-08-10
> **Response to Reviewer R8s1**
>
> Thank you very much for your feedback and support of our paper submission. We address your comments below.
>
> **Regret analysis compared to our analysis**
>
> Regret minimization results are a great suggestion and we are looking forward to studying these in more detail as part of future research.
>
> **Novelty**
>
> While we agree that the ideas in the paper extend Khetarpal et. al. (2020), the extension to options and partial option models is not straightforward and far from trivial. Specifically, the formulation of trajectory based intents for temporally extended actions is completely novel. Please see also the common response to reviewers.

---

> > ### Comment · Reviewer_R8s1 · 2021-08-26
> > **Acknowledgement**
> >
> > Thanks for the response, and good luck with the paper.

---

### Official Review · Reviewer_4o6d · 2021-07-15

**Rating:** 7
**Confidence:** 4

**Summary:**

This paper presents a formalism for semi-Markov decision processes that defines "intention" (desirable outcomes of behavior) and "affordances" (behaviors that achieve an intention). The main idea is that the agent may have many options, but may have a more limited set of intentions, and affordances can constrain the problem of selecting a good option. Theoretical results are also presented, bounding the sub-optimality caused by operating in intention/affordance/option space, rather than in primitive actions. Finally an illustrative empirical experiment gives a proof of concept for learning affordances, given fixed options and intentions.

**Limitations And Societal Impact:**

The paper does not discuss negative societal impact, but I think this work is unlikely to have significant societal impact in the foreseeable future.


**Main Review:**

---Originality---

The paper acknowledges that, conceptually, this framework owes a great deal to recent work. The extension of that existing formalism, which dealt only with primitive actions, to the SMDP case is non-trivial and, as far as I know, novel.

---Quality---

I am not aware of technical flaws in the theoretical work and the experiments seem well-designed.

---Clarity---

I found the paper to be well-written and clearly presented.

---Significance---

I think the problem that this paper addresses has the potential to be very important. One of the things that has held options back from being a complete solution for temporal abstraction is that they are defined in terms of their mechanics (i.e. a policy etc.) and it is very hard to know what option policies the agent might want to have access to. The idea of instead working in terms of things the agent wants to cause in the world seems like it could relieve pressure on option selection -- it makes it okay to have lots of options, even if a lot of them are not very helpful. It seems like it might be easier to select useful intentions than to select useful options as well -- from experience in the world the agent might be able to identify circumstances that lead to reward and that it might want to recreate.

It also seems to me that this framework is much more important in the SMDP setting than in the MDP setting, where the space of possible behaviors and possible outcomes is already quite limited.

All that said, what is presented in the paper has practical limitations, especially that the options and intentions are presumed to be given. On its own, this paper is not likely to significantly impact practice in RL. However, it may be a base upon which follow-up work is built.

---Summary of the Review---

The paper presents a sound and interesting extension to an existing theoretical framework. Though conceptually it builds on prior work, it grapples with important issues not previously addressed. I think this paper would be good to have in the literature.

---After Discussion---

Thank you to the authors for the response. After reading the other reviews and discussing with the reviewers, I still feel confident that this paper makes a solid and novel contribution to an important problem. It does also have limitations, as discussed across all of the reviews. In balance, I am comfortable with my current rating.


**Time Spent Reviewing:**

3

---

> ### Author Response · Authors · 2021-08-10
> **Response to Reviewer 4o6d**
>
> Thank you very much for your feedback and support of our paper submission. We address your comments below.
>
> **Impact and Limitations of the work**
>
> We acknowledge that intents and options are pre-specified. However, the knowledge needed for intent specification is often available, for example in user-oriented services such as web-navigation tasks or dialogue systems, or in robot manipulation. Options could be learned instead of being prespecified, but this goes beyond the scope of the paper.
>
> We characterize assumptions which are needed for intents and affordances to exhibit advantages. E.g in a low-data regime, an intermediate size of affordances (much smaller than the entire state-option space) can be optimal for planning. Similarly, the intent approximation error governs how good the estimate of the optimal option-value function can become. These insights are different when moving from actions to options. Please see also the common response to reviewers discussing the significance and impact of our work.
>
> **Learning intents \& options**
>
> To learn intents, we envisage an iterative algorithm which alternates between learning intents and affordances, so that intents are refined over time. Our analysis is complementary to any method or oracle providing intents. Additionally, since in our experiments we already learn affordances, an end-to-end framework for learning interns, options, affordances and models is not that far.

---

### Official Review · Reviewer_q4gZ · 2021-07-19

**Rating:** 5
**Confidence:** 1

**Summary:**

This paper describes options models as partial models, meaning that the options model predict only the outcome of a subset of state-action pairs (both the state transition and the reward). Options enable temporal abstraction by allowing an agent to select an option at a particular decision point and follow it until it satisfies the termination condition. The primary contribution of the paper is the development of theory which quantifies the loss incurred by using temporal abstraction (options) relative to single step decision making.



**Limitations And Societal Impact:**

Yes.

**Main Review:**

This is a clearly written paper which presents a variety of theoretical results. The majority of the work for the proofs is in the supplementary material, and the paper claims that it is somewhat distinct from Kheterpal 2020a (see line 195-196 and line 203-204). Irrespective of this point, the new bounds, to the best of my understanding, have not been demonstrated to be non-trivial. The motivation of the paper is that humans use abstraction to make better decisions, the bounds show that we can establish that plans with temporally abstract partial models are at least not unboundedly worse than standard plans. Experiments of sufficient complexity could help to motivate that empirically the methods presented here achieve the stated goal of better planning, but the 5x5 grid-worlds are just too simplistic to be convincing.

Comments:
1. In figure 3 why does the success rate of the everything-everything option set get worse with more environment interactions. The claim is that this is due to option misspecification? What does that precisely mean. Which constants would option misspecification show up in in Table 2?

2. In the experiments is everything-everything really the best baseline we can come up with? Shouldn't we compare to a non-options non-temporally-abstract method? And added bonus would be to compare the results both with unbounded computational budget and bounded.

2. On line 332-333 the paper claims that theoretical analysis suggests that it is possible to achieve faster planning across different time scales. Where is the improvement in planning speed discussed or quantified? I see bounds on planning loss and value loss, but I don't see how either directly implies that planning speed is improved. I don't see any way to directly connect the degree of abstraction/planning speed improvement with degree of intent satisfaction. Moreover, the experiments themselves don't discuss how much time is saved in a planning cycle.

**Time Spent Reviewing:**

3

---

> ### Author Response · Authors · 2021-08-10
> **Response to Reviewer q4gZ**
>
> Thank you for feedback on our submission. We address your comments below.
>
> **Novelty**
>
> While we agree that the ideas in the paper extend the work in Khetarpal et. al. (2020), the extension to options and partial option models is not straightforward and far from trivial. Specifically, the formulation of trajectory-based intents for temporally extended actions is completely novel and does not arise in the primitive case. Please also see the common response to reviewers.
>
> **Planning speed trade-offs and improvement**
>
> Re **“Where is the improvement in planning speed discussed or quantified?”.** 1. Our result on the planning loss bound suggests that there is a tradeoff between \emph{estimation} (via the model learning depending on the data size $n$) and \emph{approximation} (via the specification of intents). Note that the estimation error is a function of the number of samples (n). For a fixed amount of data, the bound suggests that the planning loss will depend on the size of the affordances and the policy class, as opposed to the size of the entire state-action space along with the corresponding policy class, which can be much larger. Hence, stronger planning loss guarantees hold when using a policy class using only affordable options. We will clarify this in text after Theorem 2.
>
> Re **Gains in sample complexity.** We also corroborate the gains in the planning speed with theoretical guarantees on sample and computational complexity of obtaining an $\varepsilon$-estimation of the optimal option value function, given only access to a generative model (See Appendix Sec. C). Due to space constraints, this result is in the appendix. We also note an error on our part: we refer to Sec C in line 332-333 without mentioning the appendix, which is perhaps the cause of confusion.
>
> Please see our response below about the experiments and the time saved in the planning.
>
> **Experiments**
> 1. **Re simplistic domain:** we reiterate that the experiments are meant to be an illustration of the theoretical concepts. The Taxi domain is commonly used to evaluate HRL-based agents (See Abel 2016, Diuk 2008, Dietterich 1999). While 3D environments from pixels are certainly more visually impressive, our environment still allows scaling in the number of state-option pairs (30k+) without needing to edit complex environment engines. We chose Taxi to do carefully controlled experiments quickly without needing to resort to the usual DeepRL tricks and tuning. This is definitely within scope of a future paper!
> 2. **Re baseline:**  The concept of temporally extended intents is not applicable to primitive actions. As a result, the options baselines here do not directly compare to the flat baseline. Moreover, our primary motivation for these experiments is to demonstrate the utility of affordance-aware option models. For this, the most relevant baseline is indeed everything-everything where model learning and planning happens over all state-option pairs.
> 3. **Re “experiments [..] don't discuss how much time is saved in a planning cycle”:** Thanks for bringing this up, we have now included a new plot in the supplementary material showing the decrease in the number of planning iterations of value iteration as the model becomes better. The summary of the plot is that the # of iterations to converge is lower for RelevantPickupDrop->RelevantPickupDrop than Everything->RelevantPickupDrop.
> 4. **Re everything->everything in Fig3:**  Initially, our intuition behind the success rate of everything-everything getting worse with more environment interactions was option misspecification. We were referring to the fact that the model requires us to learn the reward for 37500 entries and is more likely to learn a poor model for options that can easily solve the environment. To investigate this we ran subsequent experiments with lower learning rate and have found a simpler explanation: If we consider a lower learning rate, it takes the full 40M environment steps for Everything->Everything to reach a success rate ~0.9 while affordance-based training still maintains sample efficiency and perfect success rate.

---

> > ### Comment · Reviewer_q4gZ · 2021-08-30
> > **Reply**
> >
> > Thanks for the reply and additional clarification. I will update my score.

---

### Author Response · Authors · 2021-08-10
**General Response to All Reviewers**

We are encouraged by the positive reviews and thank the reviewers for their valuable insights and feedback. Below are some common comments, and we also answer specific questions in the replies to individual reviewers.

**Significance and Impact**

While our work builds on Khetarpal et al (2020), it provides a non-trivial extension from primitive actions to options and a significant step towards practicality through the formalization of partial option models. Specifically, our main contributions are:
1. Formalization of option intents and affordances.
2. Novel theoretical results highlighting the trade-offs involved in building affordance-aware temporally abstract partial models and planning with them, which can help in further algorithmic developments.
3. Experimental illustration of the theoretical results, and the evaluation of an approach for end-to-end learning of affordances combined with temporal abstraction.

**Experiments**

The reviewers raised very important questions regarding the experimental results. Here’s a summary of what we have updated and clarified:
1. We have re-run our heuristic experiments in 5.1 with lower learning rates, resulting in stabilization of the diverging agent in the Everything->Everything setting.
2. We added a new figure in the supplementary material showing the improvement in planning iterations from using affordances. We expect these results to be much more convincing, especially for Reviewer q4gZ.
3. We updated all the figures to be colorblind-friendly with less smoothing and line styles to better highlight differences between the approaches.

---

### Decision · Program_Chairs · 2021-09-27

**Decision:**

Accept (Poster)

**Comment:**

This paper introduces an option model based on the notions of intent and affordances of options, and provides a theoretical analysis of the framework. While the empirical results are limited, it was felt that the conceptual and theoretical contributions of this paper are exciting, and could be a useful contribution to the field.
There were several points of confusion raised by the reviewers, and these points should be clarified in the final version.